# Salvaging high-quality genomes of microbial species from a meromictic lake using a hybrid sequencing approach

Yu-Hsiang Chen [1,2,3], Pei-Wen Chiang[3], Denis Yu Rogozin[4,5], Andrey G. Degermendzhy[4], Hsiu-Hui Chiu[3] & Sen-Lin Tang [2,3✉]

Most of Earth's bacteria have yet to be cultivated. The metabolic and functional potentials of these uncultivated microorganisms thus remain mysterious, and the metagenome-assembled genome (MAG) approach is the most robust method for uncovering these potentials. However, MAGs discovered by conventional metagenomic assembly and binning are usually highly fragmented genomes with heterogeneous sequence contamination. In this study, we combined Illumina and Nanopore data to develop a new workflow to reconstruct 233 MAGs —six novel bacterial orders, 20 families, 66 genera, and 154 species—from Lake Shunet, a secluded meromictic lake in Siberia. With our workflow, the average N50 of reconstructed MAGs greatly increased 10–40-fold compared to when the conventional Illumina assembly and binning method were used. More importantly, six complete MAGs were recovered from our datasets. The recovery of 154 novel species MAGs from a rarely explored lake greatly expands the current bacterial genome encyclopedia.

[1] Bioinformatics Program, Taiwan International Graduate Program, National Taiwan University, Taipei, Taiwan. [2] Bioinformatics Program, Institute of Information Science, Taiwan International Graduate Program, Academia Sinica, Taipei, Taiwan. [3] Biodiversity Research Center, Academia Sinica, Taipei, Taiwan. [4] Institute of Biophysics, Siberian Branch of Russian Academy of Sciences, Krasnoyarsk, Russia. [5] Siberian Federal University, Krasnoyarsk, Russia. ✉email: sltang@gate.sinica.edu.tw

Rapid developments in bioinformatics and sequencing methods enable us to reconstruct genomes directly from environmental samples using a culture-independent whole-genome-shotgun metagenomic approach. These genomes, also called metagenome-assembled genomes (MAGs), have become a crucial information source to explore metabolic and functional potentials of uncultivated microorganisms[1–4]. Mining MAGs quickly expands our knowledge of microbial genome, diversity, phylogeny, evolution, and taxonomy[1–4]. For example, 18,365 MAGs were identified out of a total of 410,784 microorganisms in the Genomes OnLine Database (GOLD)[5]. A total of 52,515 MAGs were assembled from diverse habitats, and the MAG collection contains 12,556 potentially novel species and expands the known phylogenetic diversity in bacterial and archaeal domains by 44%[3].

Although genome-resolved metagenomics has revolutionized research in microbiology, significant challenges need to be overcome to make MAGs more accurate, reliable, and informative[1]. First, most MAGs are derived from the metagenomic assembly of short reads[1,6], and these short-read-derived MAGs usually comprise numerous short contigs rather than complete or nearly complete genomic sequences, and thus important information on genomic characters is missed, such as operons, gene order, gene synteny, and promoter/regulatory regions. As of March 2021, only 177 out of 84,768 MAGs released in NCBI were complete[7]. Second, fragmented MAGs usually miss some gene sequences and comprise unknown contaminant sequences, mistakenly assembled into the contigs[1]. Hence, low contiguity, high fragmentation, and unwanted contamination in short-read MAGs greatly affect further analyses in a variety of microbial genome-related studies.

The emergence of long-read sequencing platforms (also called third-generation sequencing platforms), such as Nanopore and PacBio, provides an opportunity to improve the contiguity of MAGs and even reconstruct complete MAGs from complex microbial communities[8,9]. Recently, researchers started to develop new assemblers to reconstruct microbial genomes with high accuracy and long contiguous fragments from long-read metagenomic datasets. In 2019, Nagarajan et al. developed a hybrid assembler called OPERA-MS[10]. The assembler yielded MAGs with 200 times higher contiguity than short-read assemblers used on human gut microbiomes. In October 2020, Pevzner et al. developed metaFlye, a long-read metagenome assembler that can produce highly accurate assemblies (approximately 99% assembly accuracy)[11,12]. The success of these newly developed assemblers becomes an important stepping-stone for reconstruction of complete MAGs with high accuracy. However, there is still much room to improve the procedures around data processing and assembling MAGs with long reads. The current study presents a new workflow for this purpose.

Our workflow combines Illumina sequencing reads and Nanopore long sequencing reads to recover many novel high-quality and high-contiguity prokaryotic MAGs from Lake Shunet, southern Siberia, one of only four meromictic lakes in all of Siberia. The lake contains stratified water layers, including a mixolimnion layer at 3.0 m, chemocline at 5.0 m, and monimimnion at 5.5 m. The mixolimnion is dominated by *Cyanobacteria*, and the chemocline contains dense and visible purple sulfur bacteria populations (>10^8 cells/mL). Our previous 16 rRNA amplicon survey showed that the lake contains diverse microbial communities with a higher Shannon diversity index and Chao1 richness estimator than Lake Shira and Lake Oigon, two another saline meromictic lakes near the center of Asia[13]. More importantly, it showed that the lake comprises at least hundreds of unknown bacteria and archaea[14], highlighting the importance of mining microbial MAGs from this rarely explored lake. However, though we attempted to recover MAGs from these layers using deep Illumina sequencing with approximately 150 Gb, only one high-quality but still fragmented MAG was obtained[14]. Hence, in this study, we developed a new workflow combining Illumina and Nanopore sequencing reads by integrating several cutting-edge bioinformatics tools to recover and reconstruct MAGs with high contiguity and accuracy. We demonstrate that our newly built workflow can be used to reconstruct hundreds of complete high-quality MAGs from environmental samples in a high-complexity microbial community.

## Results and discussion

**Reconstruction of metagenome-assembled genomes with high contiguity from Lake Shunet by combining Nanopore and Illumina sequences.** To recover novel metagenome-assembled genomes (MAGs) with high contiguity without compromising accuracy, 3.0, 5.0, and 5.5 m depth Lake Shunet samples were sequenced by Nanopore machines individually, and the resulting long reads (LRs) were analyzed together with short reads (SRs) using a workflow we developed for this study (Fig. 1a and Supplementary Table 1). Originally, we only used metaFlye, a state-of-art long-read metagenome assembler that can provide 99% accuracy[11,12], to assemble the LRs. However, recent studies found that assemblies from long reads contain numerous in-del errors, leading to erroneous predictions of open reading frames and biosynthetic gene clusters[1,11]. Incorrectly predicting open reading frames also affects the estimation of genome completeness by single-copy marker gene method, such as CheckM[15]. Hence, the contigs generated by LRs were polished using pilon[16] with SRs from Illumina sequencing. The SRs were first mapped to the assemblies from LRs with BWA-MEM[17]. Sequentially, pilon was used to automatically evaluate the read alignments to identify the disagreement between assemblies and SRs and makes corrections to fix base errors and small indels based on the evidence from alignments weighted by base coverage and quality of the SRs.

To recover more MAGs and improve contiguity, the assemblies from SRs and LRs were combined before binning. The contiguity of MAGs generated by combining two sequencing reads was dramatically higher than that from the Illumina assembly alone. The average N50 of MAGs from SRs only were 12.4, 6.0, and 7.2 kb in the 3.0, 5.0, and 5.5 m dataset, respectively. Average N50 increased to 476.5, 269.5, and 91.2 kb (Fig. 1b), respectively, when assembling with a combination of the two sequencing methods. A previous study showed that the qualities of MAGs can be improved by reassembly[18], so the step was incorporated into our workflow. When the MAGs were reassembled and selected, the average N50 increased from 476.5 kb to 530.0 kb in the 3.0 m dataset and 91.2 kb to 107.3 kb in the 5.5 m datasets (Fig. 1b).

The correlations between read coverages and contiguity were determined (Fig. 1c, d). The results revealed that the N50 values were more correlated with the Nanopore read coverage (Spearman's $r = 0.7$) than the Illumina coverage (Spearman's $r = 0.33$). This is consistent with the previous observation that contiguity plateaued when the coverage of SRs reached a certain point because the assembly of SRs cannot solve repetitive sequences[10]. Nevertheless, LRs can address the issue by spanning repetitive regions. We also found that using SR assembly only, we cannot obtain MAGs with N50 > 100kbp. By comparison, using our workflow, we can obtain 73 MAGs with N50 > 100kbp. The mean SR coverage of these MAGs was 187 times, and mean LR coverage of them was only 67. Additionally, our data size of LRs is about 1/3 that of SRs. Taken together, it represents that the contiguity of MAGs can be greatly improved with one-third LRs. The results highlighted that (1) combining two sequencing methods yield significant improvements in the qualities of MAGs that are recovered from high-complexity metagenomic datasets, and (2)

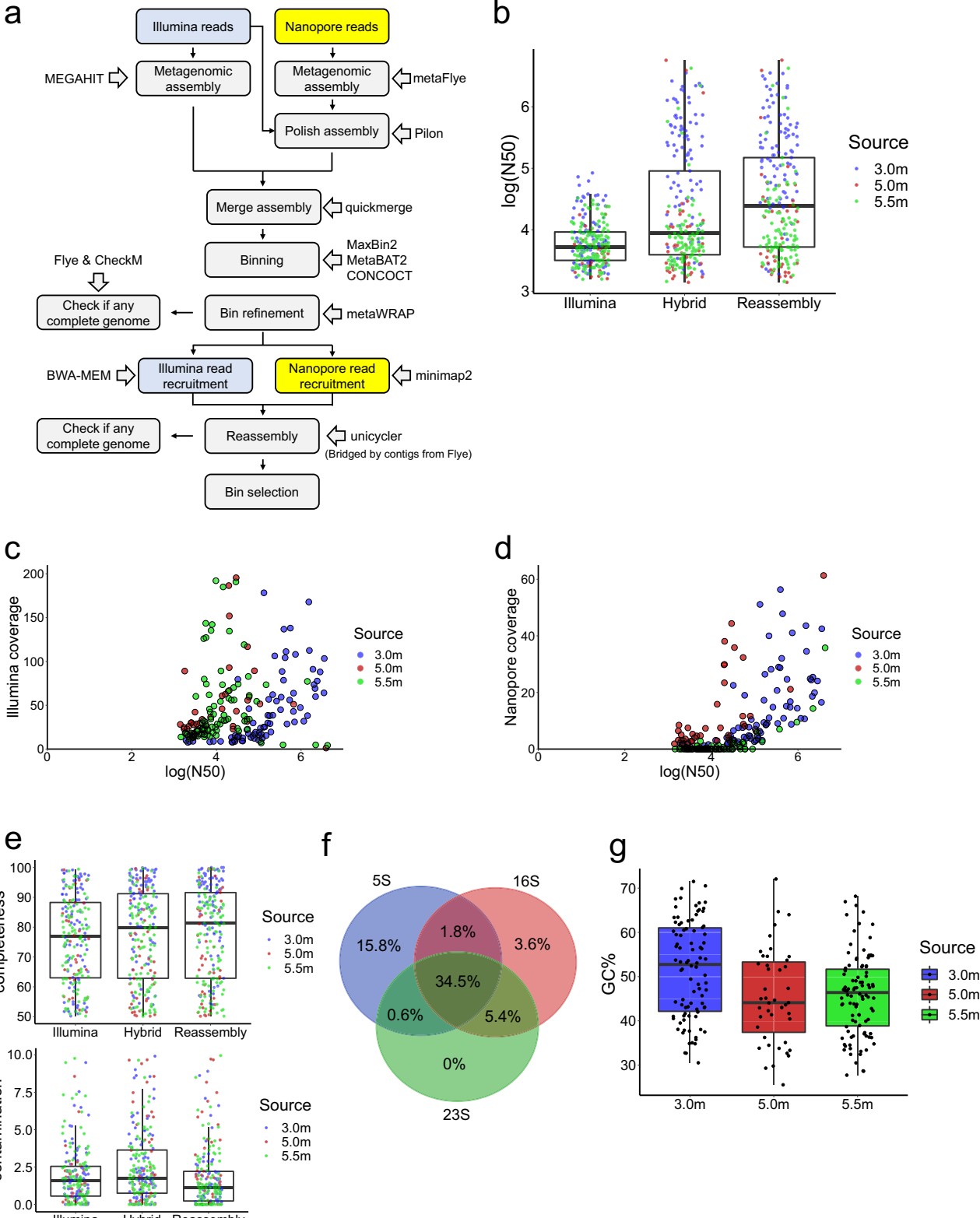

**Fig. 1 Recovery of genomes from Lake Shunet using long- and short-read sequencing. a** The workflow for assembling metagenome-assembled genomes (MAGs). **b** The value of log (N50) at 3.0, 5.0, and 5.5 m depth using SRs only, combining SRs an LRs (Hybrid), and reassembly of bins using the hybrid method. **c** Scatter plot of SR coverage and log (N50) in the final Shunet MAG collection (Reassembly). **d** Scatter plot of LR coverage and log (N50) in the final Shunet MAG collection (Reassembly). **e** The completeness and contamination of MAGs recovered from the hybrid assembly with reassembly. **f** Venn diagram from the ratio of MAGs of the hybrid assembly, containing 5 S, 16 S, and 23 S rRNA gene sequences. **g** The GC ratio of MAGs recovered from the 3.0, 5.0, and 5.5 m depth datasets. For each box plot, the center line represents median, box limits represent upper and lower quartiles, and whisker represents 1.5X interquartile range.

with only extra one-third LRs, we could retrieve genome information, such gene order, from previous SR-derived MAG collections.

Using our workflow, a total of 233 MAGs with completeness >50% and contamination <10% were reconstructed. For Genome Taxonomy Database (GTDB) species representatives, the genome quality index, defined as completeness − 5 × contamination, should be larger than 50[19,20]. To meet the GTDB standard, the MAGs were filtered by this criterion, and the MAGs with low SR coverages (<80%) were discarded, resulting in 187 MAGs (Supplementary Data 1). All the MAGs satisfied or surpassed the MIMAG standard for a medium-quality draft[21]. The median completeness of MAGs was 76.92% for SR-only and 81.26% for hybrid assembly. The median contamination was 1.61% and 1.14% for SR-only and hybrid assembly, respectively (Fig. 1e). Moreover, 45.3% of the MAGs contained 16 S rRNA gene sequences, and 34.5% of MAGs had 23 S, 16 S, and 5 S rRNA gene sequences (Fig. 1f). The median GC ratio of MAGs from 3.0, 5.0, and 5.5 m were 52.75, 44.1, and 46.4%, respectively (Fig. 1g). We also used OPERA-MS to retrieve MAGs from SRs and LRs. However, only 26 medium-quality or high-quality MAGs were recovered, indicating that the method is suboptimal in our case.

**Phylogenetic diversity and novelty of MAGs.** To explore the diversity of MAGs, we clustered and de-replicated the genomes based on a 95% ANI cutoff for bacterial species demarcation[22], since identical microbial species may be detected and assembled from the three different layers. The procedure led to 165 species-level non-redundant MAGs (Supplementary Data 1). The majority (93%) of the species-level MAGs could not be assigned to any known species after taxonomy annotation by the GTDB-Tk, revealing that a great deal of novel MAGs at the species and higher taxonomic ranks were detected (Supplementary Data 2). The novel MAGs comprised six unknown bacterial orders, 20 families, and 66 genera (Fig. 2a). The six MAGs assigned as unknown bacterial orders were all from 5.5-m dataset, which is consistent with our previous observation that water from 5.5 m deep (monimolimnion) contained more unknown bacteria based on the 16 S rRNA survey. Additionally, the novel MAGs contained novel *Cyanobacteria* and *Thiocapsa* species that were predominate in the mixolimnion and chemocline, respectively (Supplementary Data 2).

To examine the phylogenetic diversity in the novel MAGs, a phylogenomic tree was reconstructed using all these bacterial MAGs and representative bacterial genomes in GTDB (Fig. 2b). The result demonstrated that the MAGs widely span the bacterial phylogeny. The MAGs were distributed across 24 phyla, and 11 MAGs belonged to *Candidatus* Patescibacteria (also known as Candidate Phyla Radiation, CPR), so-called microbial dark matter because not enough is known about their biology[23]. Recovered MAGs also included "Margulisbacteria," "Bipolaricaulota," "Cloacimonadota," and "Caldatribacteriota," phyla that were novel and poorly characterized in the current bacterial genome databases GTDB and IMG compared to the common phyla, such as *Proteobacteria, Firmicutes, Actinobacteria, Bacteroidetes,* and *Cyanobacteria*. "Margulisbacteria" was first identified in 2016 from metagenomes of groundwater and sediment[24]. "Bipolaricaulota" (previously known as "Acetothermia") was first recovered from the metagenome of the thermophilic microbial mat community in 2012[25]. "Cloacimonadota" ("Cloacimonetes") was first described in 2008 from anaerobic digesters[26]. These phyla have not yet been cultivated, except the first "Caldatribacteriota" isolate published in 2020[27]. There are only 37 "Margulisbacteria," 21 "Bipolaricaulota," 27 "Cloacimonadota," and 19 "Caldatribacteriota" species representative genomes in

GTDB-r95. Our newly recovered MAGs of these uncultivated phyla will be useful tools for exploring these phyla.

The phylum frequencies differed between the genome collections of the standard database and the Shunet datasets (Fig. 2c). The GTDB and GEM mainly comprised *Proteobacteria*[28,29]. In contrast, in genome collections from the Shunet datasets, the phylum frequency was enriched in the *Desulfobacterota, Verrucomicrobiota, Bacteroidota,* and "Omnitrophota." The major phyla recovered in this study also differed from MAG studies from other ancient lakes. For example, a study also recovered MAGs from Siberia in Lake Baikal[30]. The major phyla of recovered MAGs are *Proteobacteria, Verrucomicrobiota,* and *Chloroflexi*. On the other hand, a 2021 metagenomics study that reconstructed MAGs from Lake Tanganyika, a freshwater meromictic lake, had higher fractions of *Proteobacteria* and *Actinobacteriota*. In both datasets, "Margulisbacteria," "Bipolaricaulota," and "Caldatribacteriota" were not seen. These results suggest that, to gain a comprehensive picture of the microbial genomes on earth, there is a strong need for future studies to explore microbiomes from various habitats, especially overlooked or understudied ones[28,31].

*Desulfobacterota* was formally proposed as a novel phylum that includes the taxa previously classified in the class *Deltaproteobacteria*[32]. Many *Desulfobacterota* are sulfate-reducing bacteria (SRB), and play importance roles in the sulfur cycle. For example, a study recovered numerous *Desulfobacterota* MAGs from a Siberian soda lake with complete cycling between sulfate and sulfide[33]. In some anaerobic aquatic systems, GSB formed syntrophic interactions with SRB via sulfur exchange, which were also observed in meromictic lakes such as Lake Faro and Ace Lake[34–36]. *Desulfobacterota* MAGs recovered in this study were from 5.0- and 5.5-m datasets. These two layers were dominated by purple sulfur bacteria (PSB) and green sulfur bacteria (GSB), respectively. The enrichment of recovered *Desulfobacterota* MAGs may be due to GSB having syntrophic interactions with diverse *Desulfobacterota* in the monimolimnion.

*Verrucomicrobiota* and "Omnitrophota" belong to the PVC group, and both were found and proposed recently. *Verrucomicrobiota* are abundant and ubiquitous in various soil and water systems. Although they have received more attention recently, only a few of them have been isolated, and their functions and ecophysiologies in water systems are not widely understood. A study in four Swedish lakes showed that the *Verrucomicrobiota* are associated with cyanobacterial blooms[37]. On the other hand, many studies showed that *Verrucomicrobia* contain higher proportions of carbohydrate-active enzymes-related genes[38,39] and can digest complex polysaccharides for growth[40]. *Verrucomicrobia* may also serve as important (poly)saccharide degraders in Lake Shunet.

**Novel predicted secondary-metabolite biosynthetic clusters and carbohydrate-active enzymes from newly recovered MAGs.** Here we demonstrate (a) the value of recovering MAGs from rarely investigated habitats to mine novel functional genes and (b) the advantage of combining SRs and LRs using two examples: secondary metabolite biosynthetic gene clusters (BGCs) and carbohydrate-active enzymes (CAZymes). Secondary metabolites are usually unique in one or a few species, and not related to the normal growth of the organisms[41]. The secondary metabolites, associated with ecological interactions, can serve as toxins, factors participating in symbiosis with other hosts, defense mechanisms[41,42]. Identifying structurally and functionally novel secondary metabolites enables us to understand the ecological interactions among the microbes. The majority of bacteria remain uncultivated, so mining novel BGCs in metagenomes provides the opportunity to discover new secondary metabolites[43,44].

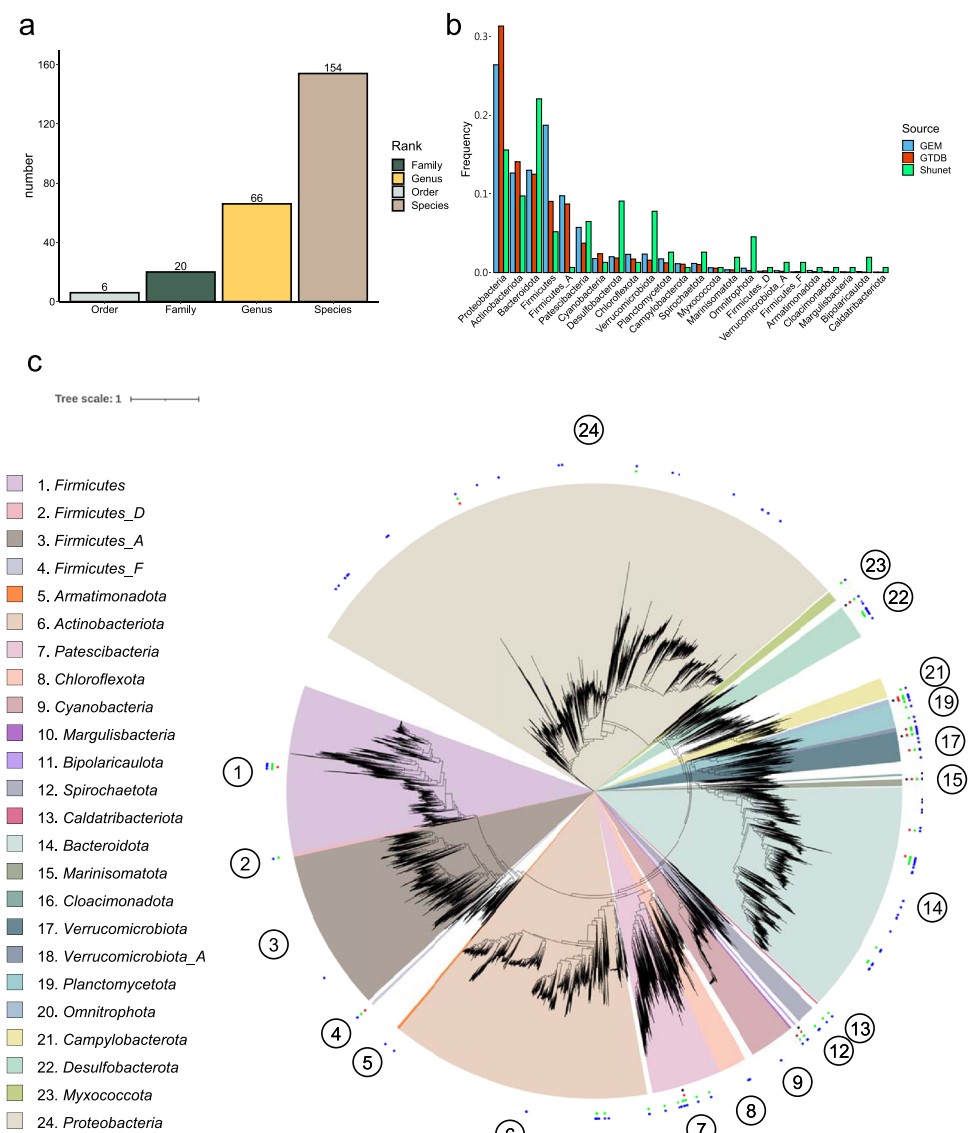

**Fig. 2 Taxonomical and molecular phylogenetic analyses of recovered bacterial MAGs. a** The numbers of novel taxonomic ranks of MAGs assigned by GTDB-Tk. **b** The phylum frequencies in the MAG collection from the Shunet dataset, GTDB representative genomes, and GEM dataset. **c** A phylogenetic tree based on the concatenation of 120 single-copy gene protein sequences. After masking, 5040 amino acid sites were used in the analysis. The phylogenetic tree includes 188 recovered bacterial MAGs and 30,238 bacterial representative genomes in GTDB-r95. The blue points represent the placement of MAGs that are classified as novel species, the green points represent novel genera, the red points represent novel families, and the black points represent novel orders. The numbers in the circle are phyla in the legend. Scale bar represents changes per amino acid site.

In our MAG collection, we identified 414 putative BGCs from 140 MAGs out of a total 165 recovered MAGs (Supplementary Fig. 1a). Among them, 134 BGCs were annotated as terpenes and 64 BGCs as bacteriocins. To determine the novelty of these BGCs, the BGCs were searched against the NCBI database using the cutoffs of 75% identity and 80% query coverage based on a previous study[28]. The results demonstrated that 384 BGCs (92%) could not be matched with these thresholds, indicating that most of these could be novel BGCs. Comparably, only 83% of BGCs were predicted to be novel BGCs in the recently published Genomes from Earth's Microbiome catalog (GEM)[28].

Complete BGCs are important because they help us identify the metabolites that these BGCs produce using molecular approaches[42]. 72% of BGCs identified from MAGs in the 3.0 m dataset were not on the edge of the contigs, suggesting that the majority of BGCs may be complete. However, only 22% of BGCs

in the 5.0 and 5.5 m datasets were not on the edges, which could be because the MAGs from 3.0 m were more contiguous because they had a 10-fold larger median N50 (Fig. 1b). In total, 213 BGCs (51%) we recovered were not on the edges. By comparison, only 34% BGCs predicted in the GEM MAG collection were not on the edge. In the 414 BGCs, 552 core biosynthetic genes, 1224 additional biosynthetic genes, 205 regulator genes, and 185 transporter genes were identified. This information will enable us to examine the products of the BGC, the function of these genes, and the roles of the products in the individual bacterium. Additionally, the results also showed that the increased contiguity of MAGs by LRs enables us to obtain more complete BGCs.

Carbohydrate-active enzymes have a range of applications. For instance, CAZymes are used for food processing and food production[45–48]. Exploring novel CAZymes in the metagenome can benefit food industries[45,46]. On the other hand, identifying

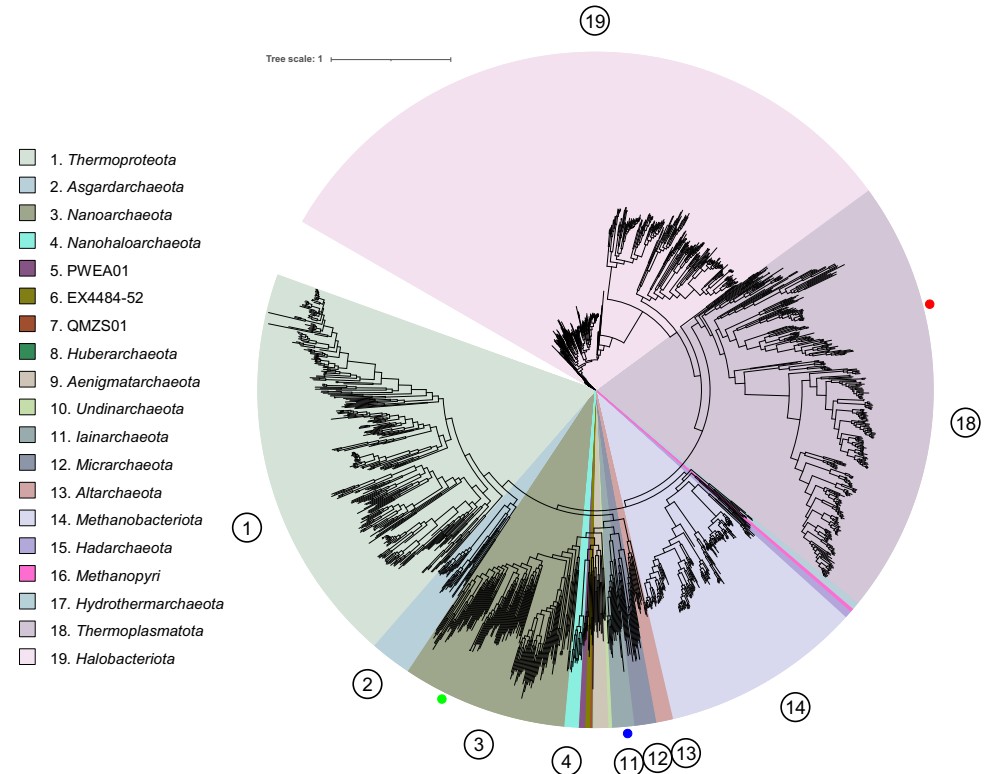

**Fig. 3 Molecular phylogenetic analysis of recovered archaeal MAGs.** The phylogenetic tree was reconstructed based on the concatenation of 122 single-copy gene protein sequences. After masking, 5124 amino acid sites were used in the analysis. The phylogenetic tree including three MAGs from Lake Shunet and 1672 archaeal representative genomes in GTDB-r95. The blue dot represents the placement of MAG M55A2, the red dot represents MAG M55A1, and the green dot represents MAG M55A3. The numbers in the circle are phyla in the legend. Scale bar represents changes per amino acid site.

novel CAZymes modules enables us to produce novel bioactive oligosaccharides that can be used to develop new drugs and supplements[47,48]. From the MAGs reconstructed in this study, we identified 8750 putative CAZymes: 3918 glycosyltransferases, 3304 glycoside hydrolases, 738 carbohydrate esterases, and 92 polysaccharide lyases (Supplementary Fig. 1b). Previous studies indicated that 60–70% protein identity can be used as a threshold for the conservation of the enzymatic function[49–51]. Among the CAZymes we identified, 1745 (44%) glycosyltransferases, 1456 (44%) glycoside hydrolases, 267 (36%) carbohydrate esterases, and 57 (62%) polysaccharide lyases shared less than 60% protein identity with their closest homologs in the NCBI nr database (Supplementary Fig. 1c). This indicates that these CAZymes could have novel carbohydrate-active functions, which future research efforts can explore further.

**Novel candidate archaeal families identified from Lake Shunet.** From the 5.5 m dataset, we identified two MAGs belonging to candidate novel families under *Methanomassiliicoccales* and *Iainarchaeales* (MAG ID: M55A1 and M55A2, respectively) based on the relative evolutionary divergence (RED) and phylogenetic placement determined by GTDB-Tk (Supplementary Data 2), and one MAG belonging to a potential novel species under *Nanoarchaeota* based on a 95% ANI cutoff for species boundary (Fig. 3 and Supplementary Data 2). In the archaeal phylogenomic tree, M55A1 formed a clade basal to the clade containing species within the *Methanomethylophilaceae* family, a group of host-associated methanogens, and the branch was supported by a 94.7% UFBoot value (probability that a clade is true) (Fig. 3)[52]. The M55A1 and *Methanomethylophilaceae*-related clade formed a superclade that is adjacent to *Methanomassiliicoccaceae*-related clade, a group of environmental methanogens[53]. These clades formed the order *Methanomassiliicoccales*, the hallmark

of which is the ability to produce methane. However, M55A1 did not contain genes encoding for a methane-producing key enzyme complex (Supplementary Fig. 2). For example, genes encoding methyl-coenzyme M reductase alpha (*mcrA*), beta (*mcrB*), and gamma subunit (*mcrG*), a key enzyme complex involved in methane production, were absent in the M55A1. On the other hand, we did not find *Methanomassiliicoccaceae*-related *mcrA*, *mcrB*, or *mcrG* genes in the other bins and unbinned sequences in the 5.5 m dataset. Furthermore, M55A1 lacks most of the core methanogenesis marker genes identified in *Methanomassiliicoccales*.

The absence of these methanogenesis marker genes implies that the archaea may have lost their methane-producing ability. If this is true, then a phylogenetic group of *Methanomassiliicoccales* may have lost the ability to perform methanogenesis after its ancestor evolved the ability to produce methane. The results not only showed the potential functional diversity in this clade but also highlighted how much such a little-studied environment can reveal about functional diversity in known microbial lineages.

**Five complete MAGs of a candidate novel genus and species from Lake Shunet.** The assemblies of Shunet datasets yielded six complete and circulated bacterial genomes (Fig. 4). Among these six complete MAG, two belonged to a novel *Simkaniaceae* genus, and the other four were classified as novel *Cyanobium* species, *Thiocapsa* species, and species under GCA-2401735 (an uncharacterized genus defined previously based on phylogeny), according to the GTDB taxonomy inference based on ANI and phylogenomic analyses (Supplementary Data 1 and 2). The following are individual descriptions of their unique taxonomic and metabolic features. The nitrogen, carbon, sulfur, and energy metabolisms are described in Supplementary Fig. 3.

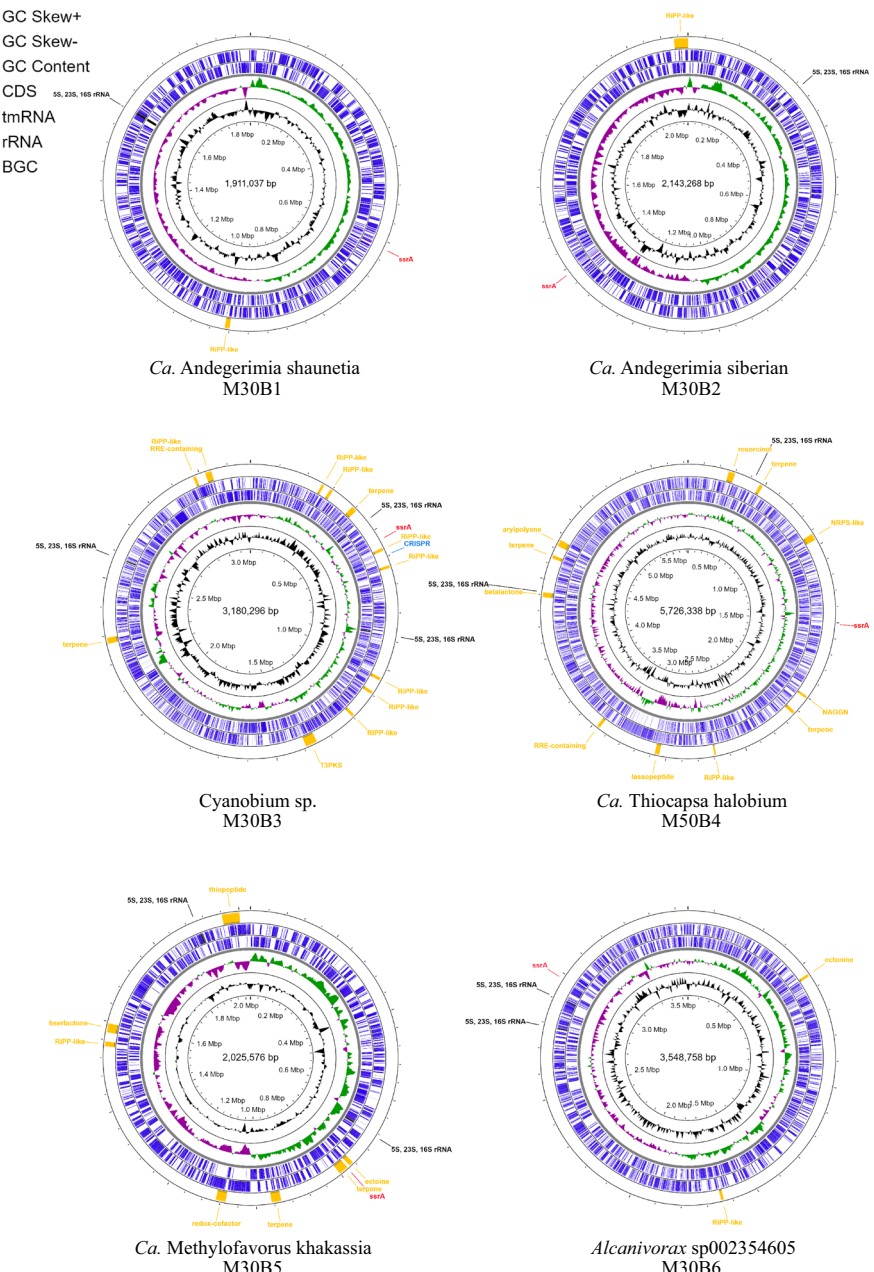

**Fig. 4 Representation of the six complete MAGs.** The rings from the inside to outside represent GC content (black), GC skew- (purple), and GC skew + (green). The next two blue rings represent coding sequences on the forward and reverse strands, respectively. On the outmost ring are the rRNA gene sequences (black), transfer-messenger RNA (red), and secondary metabolite gene clusters (yellow). MAG ID M30B6 is classified as *Alcanivorax* sp002354605, M30B1 and M30B2 are *Simkaniaceae* sp., M30B3 is *Cyanobium* sp., M30B5 is "Methylofavorus khakassia"., and M50B4 is "Thiocapsa halobium".

A candidate novel *Simkaniaceae* genus was identified. According to GTDB-TK, there were two complete MAGs—M30B1 and M30B2—assigned as an unclassified genus under *Simkaniaceae*, a family in the class *Chlamydiia*, based on the topology of the phylogenetic tree. M30B1 and M30B2 formed a monophyletic group and shared 72.48% percentage of conserved protein (PCOP), above the genus boundary of 50% PCOP[54]. The genomes shared 77% ANI, below the 95% cutoff for the same species[22], and the identity of their rRNA gene sequences was 98.45%. Together, the results showed that the two MAGs were two different new species under a novel genus. Therefore, we propose a new genus, *Candidatus* Andegerimia, to include the two MAGs, and renamed the two MAGs as *Candidatus*

Andegerimia shunetia M30B1 and *Candidatus* Andegerimia siberian M30B2, abbreviated as M30B1 and M30B2, respectively.

*Simkaniaceae*, like all *Chlamydia*, are obligately intracellular bacteria that live in eukaryotic cells[55]. Validated natural hosts include various multicellular eukaryotic organisms like vertebrates. That some *Simkaniaceae* PCR clones were identified from drinking water implies that *Simkaniaceae* may also live in unicellular eukaryotes[56]. Our samples were collected from saline water. 10-μm plankton nets were used to filter large organisms. Hence, *Ca*. A. shunetia and *Ca*. A. siberian may be derived from tiny or unicellular eukaryotic organisms.

Leveraging long- and short-read sequencing together to improve MAG reconstruction can result in comprehensive

**Table 1 KEGG orthologues that are present in the novel MAGs but absent in their sister taxa.**

| KO number | Definition |
| --- | --- |
| *Simkaniaceae* | |
| K00954 | Pantetheine-phosphate adenylyltransferase |
| K01580 | Glutamate decarboxylase |
| K15736 | L-2-hydroxyglutarate |
| K01607 | Carboxymuconolactone decarboxylase |
| K03704 | Cold shock protein |
| *Thiocapsa* | |
| K07306 | Anaerobic DMSO reductase subunit A |
| K07307 | Anaerobic DMSO reductase subunit B |
| K01575 | Acetolactate decarboxylase |
| K13730 | Internalin A |
| K05793 | Tellurite resistance protein TerB |
| K05794 | Tellurite resistance protein TerC |
| K05791 | Tellurium resistance protein TerZ |
| *Cyanobium* | |
| K07012 | CRISPR-associated endonuclease/helicase |
| K07475 | Cas3 |
| K15342 | CRISP-associated protein Cas1 |
| K19046 | CRISPR system cascade subunit CasB |
| K19123 | CRISPR system cascade subunit CasA |
| K19124 | CRISPR system cascade subunit CasC |
| K19125 | CRISPR system cascade subunit CasD |
| K19126 | CRISPR system cascade subunit CasE |

genomes with less genome contamination and fewer binning errors. The two *Simkaniaceae* MAGs we recovered contained five KEGG orthologues that were not present in known *Simkaniaceae* genomes (Table 1). First, the genomes have cold shock protein genes, and the genes were highly conserved (93% amino acid identity) between the two *Simkaniaceae* genomes. Cold shock proteins are used to deal with the sudden drop in temperature[57]. The proteins are thought to be able to bind with nucleic acids to prevent the disruption of mRNA transcription and protein translation caused by the formation of mRNA secondary structures due to low temperature[57]. The water temperature of samples in the mixolimnion we collected in July 2010 ranges from 6.5 to 15.5 °C[13,14]. In winter, ice cover the lake surface, and the temperature is below 0 °C down to 4 m deep[58]. The existence of the genes in the genomes may confer cold resistance on the *Simkaniaceae* bacteria in the mixolimnion of Lake Shunet, allowing them to withstand cold environments or rapid temperature change.

Besides the cold shock protein genes, the two *Simkaniaceae* also had glutamate decarboxylase (GAD) genes. GAD is an enzyme that catalyzes the conversion of glutamate into γ-aminobutyric acid (GABA) and carbon dioxide. Many bacteria can utilize the GAD system to tolerate acidic stress by consuming protons during a chemical reaction[59]. The system usually accompanies glutamate/GABA antiporters, responsible for coupling the efflux of GABA and influx of glutamate. The antiporter can also be found in the two novel *Simkaniaceae* genomes, indicating that the bacteria can use the system to tolerate acidic external or host intracellular environments.

Along with the unique features in the genus, we identified a difference between the two MAGs in terms of metabolism. Taking sulfur metabolism as an example, the M30B2 had all the genes for assimilatory sulfate reduction (ASR), except for *cysH*, and contained the sulfate permease gene (Supplementary Fig. 3). On the contrary, M30B1 did not contain ASR or the sulfate permease gene. This indicates that M30B2 can take up and use sulfate as a sulfur source, but M30B1 cannot.

The MAG M30B3 was classified as a novel *Cyanobacteria* species genome under the genus *Cyanobium*, based on phylogenomic tree and 84.28% ANI shared with the Cyanobium_A sp007135755 genome (GCA_007135755.1), its closest phylogenetic neighbor. We named the genome *Candidatus* Cyanobium sp. M30B3, abbreviated as M30B3. The M30B3 is the predominant bacterium in Lake Shunet at 3.0 m and plays a pivotal role in providing organic carbon in the lake ecosystem[14].

Our analysis of the M30B3 genome revealed that the bacterium harbors an anti-phage system that its known relatives lack. In the novel cyanobacterial genome under the *Cyanobium* genus, we found that the genome harbored several CRISPR-associated (Cas) protein genes that were not in other *Cyanobium* genomes (Table 1). The CRISPR-Cas system is a prokaryotic immune system that enables prokaryotic cells to defend against phages[60]. The system can be classified into six types and several subtypes according to protein content. The signature protein of type I is Cas3, which has endonuclease and helicase activities[60]. *cas3* genes can be found in the novel *Cyanobium* genome but not in other known *Cyanobium* genomes. Furthermore, the genome also had *cse1* and *cse2* genes, signature protein genes for the I-E subtype. Our results show that the novel genome harbors a type I-E CRISPR system and that this system is absent in its phylogenetic-close relatives.

Lake Shunet features dense purple sulfur bacteria (PSB) in its chemocline (5.0 m) layer (>$10^8$ cells/mL)[14], and the density is comparable to that of Lake Mahoney (Canada), renowned for containing the most PSB of any lake in the world ($4 \times 10^8$ cells/mL)[61]. A complete MAG of *Thiocapsa* species, the predominant PSB in the 5.0 m layer, was recovered from the 5.0 m dataset. The MAG was classified as a candidate novel species because it shared 90.71% ANI with the genome of *Thiocapsa rosea*. Therefore, we propose the creation of a new species, *Candidatus* Thiocapsa halobium, abbreviated as M50B4.

The complete genome of the predominant PSB M50B4 will help us understand carbon, nitrogen, and sulfur cycling in Lake Shunet. *Thiocapsa* can perform photosynthesis by reducing sulfur as an electron donor, and *Thiocapsa* can fix nitrogen[62,63]. M50B4 contained genes for bacteriochlorophyll synthesis and the Calvin cycle for carbon fixation. A previous study revealed that *Thioflavicoccus mobilis*, a bacterium close to *Thiocapsa*, can utilize rTCA and the Calvin cycle to fix carbon[64]. In M50B4, all genes for the reverse TCA cycle (rTCA), except for the ATP citrate lyase gene, were identified. Whether the M50B4 can use both rTCA and the Calvin cycle like *T. mobilis* needs to be determined. For sulfur metabolism, the MAG carried intact gene sets involved in SOX system dissimilatory sulfate reduction/oxidation. The sulfate importer gene was also seen in the MAG, which equipped the bacterium with the ability to import extracellular thiosulfate and sulfate and to use them as sulfur sources. In terms of nitrogen metabolism, like other *Thiocapsa*, the bacterium had a gene cluster to conduct nitrogen fixation and a urea transporter and urease gene cluster to utilize urea. Besides nitrogen fixation, currently available *Thiocapsa* have all genes for denitrification, and some *Thiocapsa* have genes to convert nitrite to nitrate. However, the genes were not seen in our MAG.

There are currently five cultured *Thiocapsa* species. Two of these, *T. rosea* and *T. pendens*, contain gas vesicles. Our genomic analysis revealed that M50B4 contained two copies of the gas vesicle structure protein gene *gvpA*. The gene is also present in *T. rosea* and *T. pendens*, but not in any other *Thiocapsa* genomes, indicating that the genes are critical for vesicles to exist in *Thiocapsa*, and therefore the novel species have gas vesicles. Gas vesicles enable *T.* sp. M50B4 cells to modulate their buoyancy so they can move to the locations with optimal conditions for anoxygenic photosynthesis[65]. For example, the position of the

chemocline, a zone with a $H_2S$ concentration and illumination that are optimal for anoxygenic photosynthesis, fluctuated between 4.5 and 5.5 m deep from 2003 to 2009[58]. During the period, the corresponding change in the PSB peak location was also observed, indicating that gas vesicles may help M50B4 move along with the location change in the chemocline[58]. On the other hand, 16 S rRNA analysis showed that M50B4 can also be found in Lake Shira, a less stable meromictic lake near Lake Shunet. In summer 2004, the redox zone of Lake Shira was recorded to have shifted upwards in two days[66]. This showed that the PSB are able to maintain their position near the zone by shifting upwards, indicating that gas vesicles may help M50B4 deal with rapid environmental changes.

We found that the novel *Thiocapsa* complete MAG have genes that encode dimethyl sulfoxide (DMSO) reductase subunits A and B (Supplementary Table 2). DMSO reductase is an enzyme that catalyzes the reduction of DMSO into dimethyl sulfide (DMS). The reductase enables bacteria to use DMSO as terminal electron acceptors instead of oxygen during cellular respiration[67]. The DMSO reduction reaction could impact the environment. DMS, the product of the reaction, can be emitted into the atmosphere and be oxidized into sulfuric acid[68]. Sulfuric acid can act as a cloud condensation nucleus and leads to cloud formation, blocking radiation from the sun. The flux of the anti-greenhouse gas DMS is mainly investigated and discussed in oceanic environments[69,70]. The flux and role of DMS in lake ecosystems are overlooked and rarely documented[71]. Our finding that the dense PSB in Lake Shunet carried the genes for DMS metabolism shows the need to investigate the impact and importance of DMS from bacteria in lake ecosystems and sulfur cycling.

A complete MAG, named M30B5, was classified as a novel *Methylophilaceae* species under a genus-level lineage, called GCA-2401735, which was defined based on phylogenetic placement[20]. The GCA-2401735 lineage currently only comprises two genomes —GCA-2401735 sp006844635 and GCA-2401735 sp002401735— neither of which meet high-quality genome standards due to their low completeness and lack of 16 S rRNA gene sequence. The novel complete genome can serve as a representative species of the genus and can be used to infer the capability of the genus (Supplementary Table 2). Here, we propose the genus *Candidatus* Methylofavorus to include the three GCA-2401735 genomes, and the M30B5 was renamed as *Candidatus* Methylofavorus khakassia.

The isolation locations of the three genomes imply that their habitats were distinct from those of other *Methylophilaceae*. The three "Methylofavorus" genomes were isolated from a cold subseafloor aquifer, shallow marine methane seep, and saline lake, indicating that the bacteria can live in saline environments. By comparison, most other *Methylophilaceae* members live in soil and freshwater or are associated with plants (except for the OM43 lineage)[72]. This indicates that the ancestor of "Methylofavorus" gained the ability to live in saline habitats and diverged from the ancestor of the genus *Methylophilus*, its closest phylogenetic relatives.

The complete genome of M30B5 enables us to comprehensively study metabolic potentials. *Methylophilaceae* is a family of *Proteobacteria* that can use methylamine or methanol as carbon or energy sources[72,73]. In our analysis, methanol dehydrogenase gene existed in our genome, and methylamine dehydrogenase gene was absent, indicating that the bacteria use methanol as a carbon source instead of methylamine. For motility, flagella are found in some *Methylophilaceae*. Interestingly, flagella- and chemotaxis-related genes were not identified in the MAG but were identified in the other two "Methylofavorus" species, suggesting that M30B5 lacks mobility comparing to the other two "Methylofavorus" species (Supplementary Fig. 4).

The comparative analysis of M30B5 and other "Methylofavorus" species revealed that the bacteria use different types of machinery to obtain nitrogen (Supplementary Fig. 4). The formamidase, urease, and urea transporters were present in M30B5 but not the other two "Methylofavorus" species. Instead, the two "Methylofavorus" species had nitrite reductase, which was not in our MAG. The results indicate that M30B5 can convert formamide into ammonia and formate, and take up extracellular urea as a nitrogen source. On the contrary, the other two "Methylofavorus" can use nitrite as nitrogen resources. Our analysis revealed that "Methylofavorus" is metabolically heterogeneous.

## Conclusions

In this study, we successfully developed a workflow to recover MAGs by combining SRs and LRs. This workflow reconstructed hundreds of high-quality and six complete MAGs—including six candidate novel bacterial orders, 20 families, 66 genera, and 154 species—from water samples of Lake Shunet, a meromictic lake with a diverse microbial community. It demonstrates that with extra less LRs, we can salvage important genome information from previous SR metagenomes. Using comparative genomics, unique and intriguing metabolic features are identified in these complete MAGs, including two predominant novel species: *Thiocapsa* sp, and *Cyanobium* sp[14]. The findings show that it is advantageous to apply this method in studies of microbial ecology and microbial genomics by revising and improving the shortcomings of SRs-based metagenomes. Additionally, we show that the MAGs contain a high proportion of potential novel BGCs and CAZymes, which can be valuable resources to validate and examine the metabolic flexibility of various microbial lineages through further experimental approaches and comparative genomics. Finally, this study found a high ratio of poorly detectable taxa in the public databases, suggesting that the investigation into rarely explored environments is necessary to populate the genomic encyclopedia of the microbial world, explore microbial metabolic diversity, and fill the missing gaps in microbial evolution.

## Methods

**Sample collection and information**. Water samples at 3.0, 5.0, and 5.5 m depth were collected from Lake Shunet (54° 25'N, 90° 13'E) on July 21, 2010. The collection procedure was described in our previous research[14]. 20 L of water were pumped from each depth into four sterile 5-L polypropylene bottles. Water was filtered through 10-μm plankton net to remove large particles. The water was stored for 3 h in the bottles, and part of the water was then transferred into sterile 2.0-ml screw tubes (SSIbio®) and stored at −80 °C until DNA extraction. The rest of water was concentrated using tangential flow filtration (TFF) system (Millipore) with 0.22-μm polycarbonate membrane filters. The bacteria in the retentate were then retained on cellulose acetate membranes (0.2 μm pore size; ADVANTEC, Tokyo, Japan) and stored at −80 °C until DNA extraction. The physicochemical properties of samples were mentioned in a previous study[14]. The pH was 8.1, 7.6, and 6.7; water temperature was 15.5, 9.5, and 7.5 °C; and salinity was 26, 40, and 71 g/L in the 3.0-, 5.0-, and 5.5-m samples, respectively.

**DNA extraction and sequencing**. Reads from Illumina and Nanopore sequencing platforms were used in this study. The sequencing reads from Illumina were described in our previous study[14] (Supplementary Table 1). DNA for Illumina sequencing was extracted from a TFF-concentrated sample using the cetyl-trimethylammonium bromide (CTAB) method[74]. In terms of Nanopore sequencing for 3.0-m samples, the same DNA batch used for Illumina sequencing of 3.0-m was sent to Health GeneTech Corp. (Taiwan) for Nanopore sequencing. For 5.0- and 5.5-m samples, there was no DNA remaining after Illumina sequencing, so in 2020 the DNA was extracted again from frozen water samples using the CTAB method by retaining the bacteria on cellulose acetate membranes without TFF concentration. The amounts of DNA were still insufficient for Nanopore sequencing, so the DNA samples were mixed with the DNA of a known bacterium, *Endozoicomonas* isolate, at a 1:2 ratio. No *Endozoicomonas* was detected in the water samples according to our 16 S rRNA amplicon survey[14]. The mixed DNA was then sent to the NGS High Throughput Genomics Core at Biodiversity Research Center, Academia Sinica for Nanopore sequencing. To remove reads that had originated from the *Endozoicomonas* isolate, Kaiju web server[75] and Kraken 2[76] were used to assign the taxonomy for each read; reads that were classified as

*Endozoicomonas* by Kaiju or Kraken were removed from our sequencing results. The Nanopore sequencing and processing yielded 13.83, 12.57, and 4.79 Gbp of reads from the 3.0, 5.0, and 5.5 m samples, respectively (Supplementary Table 1).

**Metagenome assembly**. MAG assembly was performed by combining short reads (SRs) from Illumina sequencing and long reads (LRs) from Nanopore sequencing; this workflow is described in Fig. 1a. First, the LRs from 3.0, 5.0, and 5.5 m datasets were individually assembled by metaFlye v2.8[12] with default settings, and the assemblies were polished with corresponding SRs using Pilon v1.23[16]. On the other hand, SRs were also assembled by MEGAHIT v1.2.9 with k-mer of 21, 31, 41, and 51[77]. The assemblies from SRs and LRs were then merged by quickmerge v0.3 with parameters -ml 7500 -c 3 -hco 8[78]. The merge assemblies were then binned using MaxBin2[79], MetaBAT2[80], and CONCOCT[81] in metaWRAP v1.3[18]. The bins from the three bin sets were then refined by the bin refinement module in metaWRAP v1.3. The resulting bins were then polished again by Pilon v1.23 five times. To reassemble the bin, sorted reads that belonged to individual bin were extracted by BWA-MEM v0.7.17[17] for SRs and by minimap2 for LRs[82]. The extracted long reads were assembled by Flye v2.8, or metaFlye v2.8 if the assembly failed using Flye v2.8[12,83]. The bins were then reassembled individually using Unicycler v0.4.8 using the extracted reads and reassembled long-read contigs which were used as bridges[84]. To determine whether the original or reassembled bin was better, the bin with higher value of genome completeness—2.5 × contamination, estimated by CheckM v1.1.3[15], was chosen and retained. Contigs labeled as circular by Flye or metaFlye, >2.0 Mb in size, and completeness >95% were considered "complete" MAGs. The complete MAGs were visualized using CGView Server[85].

While we were preparing this manuscript, Van Damme et al. published a hybrid assembler, called MUFFIN, that also integrates metaFlye and metaWRAP to recover MAGs and Unicycler for reassembly[86]. However, our workflow has a step to merge the assemblies from SRs and LRs to increase the contiguity and assembly size. Moreover, for the reassembly, we use contigs from metaFlye, instead of default setting: miniasm, as the bridge, which we found can produce a better quality reassembly.

**Annotation of metagenome-assembly genomes**. The completeness, contamination, and other statistics on metagenome-assembled genomes (MAGs) were evaluated using CheckM v1.1.3[15]. The genome statistics were processed in R[87] and visualized using the ggplot2 package[88]. The taxonomy of MAGs was inferred by GTDB-Tk v1.3.0[19]. Average Nucleotide Identities (ANIs) between MAGs were determined by FastANI v1.32[22]. MAGs were annotated using Prokka v1.14.5 with 'rfam' options[89]. To annotate MAGs with KEGG functional orthologs (K numbers), putative protein sequences predicted by Prodigal v2.6.3[90] were annotated using EnrichM v0.6.0[91]. The K number annotation results were then used to reconstruct the transporter systems and metabolic pathways using KEGG mapper[92], and the completeness of KEGG modules was evaluated using EnrichM. Secondary metabolite biosynthetic gene clusters in each MAG were identified using antiSMASH v5.0[93]. Ribosomal RNA sequences were inferred by barrnap v0.9[94].

Carbohydrate-Active Enzymes (CAZymes) in MAGs were predicted and classified using run_dbcan v2 by HMMER search against the dbCAN HMM (hidden Markov model) v9 database[95]. The putative CAZymes were blasted against the NCBI nr database using BLASTp[96] to search their closest homologs and determine the protein identities between predicted CAZymes and their homologs.

**Taxonomy assignment and phylogenetic analysis**. Taxonomic assignment was performed by GTDB-Tk v1.3.0[19]. GTDB-Tk classifies bacterial and archaeal genomes and identifies novel taxa by determining the phylogenetic placement and relative evolutionary divergence (RED) values of query genomes in the GTDB reference tree, using a 95% ANI cutoff for species boundary. Bacterial and archaeal phylogenomic trees were inferred by a de novo workflow in GTDB-Tk v1.3.0. All species-level non-redundant MAGs recovered in this study were analyzed together with the reference genomes in Genome Taxonomy Database (GTDB)[20]. In the de novo workflow, marker genes in each genome were identified using HMMER 3.1b2[97]. Multiple sequence alignments based on the bacterial or archaeal marker sets were then generated and masked with default settings. Trees of bacteria and archaea were then inferred from the masked multiple sequence alignment using FastTree with the WAG + GAMMA models and 1000 bootstraps[98]. The trees were visualized with the interactive Tree of Life (iTOL) v4[99].

**Reporting summary**. Further information on research design is available in the Nature Research Reporting Summary linked to this article.

## Data availability

All sequencing data and assembled genomes are available through National Center for Biotechnology Information (NCBI) repositories under BioProject ID: PRJNA721826. Sequence reads of metagenomes from samples at 3.0, 5.0, and 5.5 m deep can be found under SRA accession numbers SRR14300307, SRR14300308, SRR14300309, SRR14307495, SRR14307795, and SRR14307796. The accession numbers of MAGs can be found in Supplementary Data 1 and 2.

## Code availability

The workflow based on snakemake is available at https://github.com/SeanChenHCY/metaLAS.

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

## Acknowledgements

This work was supported by the Ministry of Science and Technology in Taiwan through the Taiwan–Russia Joint Project Grant NSC 102-2923-B-001-004-MY3 and Russian Foundation for Basic Research Grant 21-54-52001 and MOST 105-2923-B-001-001-MY3. Y.H.C. would like to acknowledge the Taiwan International Graduate Program (TIGP) for its fellowship towards his graduate studies. We would like to thank Noah Last of Third Draft Editing for his English language editing.

## Author contributions

Y.H.C. and S.L.T. conceived the idea for this study. Y.H.C. assembled the genomes, performed the bioinformatics analysis, and wrote the manuscript. P.W.C. and H.H.C. prepared the DNA samples. D.R. and A.D. collected water samples. S.L.T. supervised the overall study.

## Competing interests

The authors declare no competing interests.
