## [Transparent Peer Review File · Communications Biology]

Reviewers' comments:

Reviewer #1 (Remarks to the Author):

This paper describes the discovery of candidate novel microbial species through both complete or high-quality MAGs (metagenome-assembled genomes). This use of a workflow combining both short and long-read sequencing data allowed this discovery.

Overall, the paper is intelligible, well written, thorough, and describes stunning results and sturdy methodology. This workflow possesses great potential for analysis in many labs working on microbial ecology and genomics and should not be restricted to lake microbial analysis.

Due to the overall quality of the paper, I do not think Major revisions are needed.

Major comments:

The study design and the analysis done are more than satisfactory.

The results are precise and provide a glimpse of the future analysis done in microbial genomics.

I am aware of some limitations regarding the workflow, and they are well explained in the "Material and Methods" section. The restrictions I am talking about concern the reassembly step that can worsen the bin quality and the use of N50 as a sole quality indicator.

It is good to see them taken into account so clearly.

Minor comments:

In the "Results and Discussion," you mention the improvement of the N50 by the use of both LRs and SRs instead of only the SRs for the assembly. Did you analyze the change in the completeness and contamination between the two methods? If so, you might want to include it.

The workflow presented here is brilliant and has potentials, but only a few people will be tempted to follow it in the current state.

It has been studied that reproducing a paper or reproducing the workflow presented in a paper is difficult and time-consuming (<https://doi.org/10.1371/journal.pone.0080278>, <https://doi.org/10.1371/journal.pbio.3000333>).

Have you considered making a container (Docker, Singularity, Podman) of all the tools needed for reproducing the analysis?

Or have you considered making an automated pipeline with Nextflow or Snakemake that will handle the user's installation and intermediary file?

Making such an automated pipeline will increase the reproducibility of your work and the usability of the pipeline for other analyses by searchers. Such a pipeline would be a fantastic addition to the realm of tools currently available for "hybrid" metagenomics analysis.

You already provide the list of tools, versions, and parameters you used. In my opinion, it is not a requirement for the publication of this paper, but providing such an automated pipeline in the future could contribute to better and more reproducible science.

In Conclusion, this paper is an excellent paper providing exciting results on the microbial population of a meromictic lake in Siberia and a high potential workflow.

Reviewer #2 (Remarks to the Author):

General comments:

In this paper, the authors show a proof-of-concept of how short-reads based (Illumina) and long-read based (Nanopore) sequencing technology can be combined to retrieve a high number of MAGS, including "novel" ones, using their hybrid assembly method. The method is applied to samples from Lake Shunet, a meromictic lake in Siberia. The major advantage of this hybrid method is getting higher N50, as well as completely circular MAGs, which they recovered 6 of.

Overall, the paper describes the method well, although in my major comments below, I address ways that the authors can support their claims further. In some places, the claims are insufficiently supported. Some methods are missing, which makes the reading of the paper at times unclear. I have provided some specific areas for clarification below.

The major comment is to put the results in the greater literature context. There is sufficient body of literature on both long-reads sequencing in recovering MAGs, and in MAGs from meromictic lakes and stratified lakes. Doing so would make it clearer how their study is important, and what does it contribute. It would also enable a deeper discussion of the relevance of these “novel” groups.

Major comments

o Line 183-42: Genome frequencies might not be the best indicator of whether the Sunet MAGs are unique and novel, since the GTDB database comprises MAGs from many different environments. It can be hypothesized that comparing MAGs from two distinct environments, one will encounter different MAGs. A more useful comparison in terms of MAGs would be to other water lakes that encounter seasonal or annual stratification, or lakes that encounter similar environmental characteristics. For example, Cabello-Yeves et al., 2020 (ASLO L&O) has MAGs also from Siberia in Lake Baikal, and Tran et al., 2021 (ISME) has MAGs from the meromictic lake Tanganyika. *Desulfobacterota* is common in hypolimnions, as are *Verrucomicrobia*, *Bacteroidota* in lakes. The authors could put into context the MAGs that they find versus the frequency that those taxa are found in similar metagenomic datasets for example, in addition to the already existing GTDB-tk comparison.

Overall, the points about MAG novelty would be made stronger by comparing to other datasets. For example, the GEM dataset (which was mentioned in their introduction) (~52 000 MAGs) (Nayfach et al., 2020, Nature Biotechnology) could also be used to calculate the frequency of the taxa too, to see if the results about the higher proportion of X taxa remains valid.

o Line 197-98: There is substantial literature about microbial ecology on stratified lakes, including meromictic lakes, so I don't believe this statement is accurate. Overall, a more thorough screening of the water microbial ecology would allow the arguments in the paper to be put into their ecological context.

Minor comments:

o Line 80 : ref needed

o Line 109: using different binning methods on the short reads set, are more high-quality MAGs recovered? Using multiple binning softwares, followed by dereplication of the MAGs, can often increase the number of MAGs and the MAG quality. Additionally, for novel bacteria/archaea belonging to the DPANN or CPR, usual core-gene set such as those in CheckM might not capture their true completeness.

o Line 128: More methodological details are needed to describe how the SRs were used to correct the contigs generated by the LR method, such that would enable the reader to replicate their experiment.

o Line 156: GTDB / GTDB-tk requires a citation.

o Line 170: De-replicated instead of de-duplicated

o Line 182: Poorly and unusually characterized phyla – in comparison to what? Is it in comparison to other water lakes, or else?

o Line 204: What makes these metabolites novel?

o Line 208: Where are the 140 MAGs from? On line 173, the previous mention of the number of MAGs was 165.

o Line 227: Instead of “on the other hand”, “additionally” might be better to use because it would Line 242: Overuse of the novelty aspect but without support.

o Line 247: What criteria were used to determine novelty? The authors mention bootstrap support values. I see that %ANI to nearby genomes is provided for other groups later on in the manuscript, but it would be helpful to provide it here too.

- o Line 268-269: Have these Methanomassiliicoccales been found in other lakes?
- o Line 297: What membrane pore size was used?
- o Line 300: Contamination can be a result of binning, but is not always the case using binning. Additionally, it is possible (although not common) to retrieve fully circular MAGS from binning using short-reads. I suggest this phrasing to be more accurate (suggestions only, authors can reword as they see fit): "Leveraging long-reads sequencing along with short-read sequencing to improve MAG reconstruction, can result in comprehensive genomes with less genome contamination and binning errors".
- o Line 309: What are these "extreme" cold environments? Please mention the temperature, general climate, salinity levels.
- o Line 317: What is the pH in this environment?
- o Line 318-324: The argument here is that the metabolism of these two mags differs, but the only difference listed is the assimilatory sulfate reduction pathway, and the sulfate permease gene. How do other aspects of the sulfur cycle (the authors' example) differ across genomes? Do they differ in other aspects of metabolism than sulfur cycling? More evidence is needed to support that it "broadens our knowledge of the metabolic versatility in Simkaniaceae" (example: Line 324, 351).
- o Line 325 and elsewhere: Do the abbreviated MAG names (e.g. M30B3) signify anything? (e.g. sample provenance, metadata, etc.).
- o Line 370-371: Which vesicle protein genes were found?
- o Line 375: Provide clarification of how the environmental conditions are changing, and how the adaptations are possibly related to these conditions. For example, are there big changes in terms of light or oxygen, as cited in line 374-375?
- o Line 388: While the organisms carry the genes, they may not utilize them year-round. Perhaps change for "carry the genes for DMS metabolism".
- o Line 388: What is meant by "extremely dense"? What criteria is used? Also check and remove superfluous words such as "extreme" (line 388, 87, 309, 344) when no further explanations of how or why they are extreme are provided.
- o Line 432: how is microbial community complexity measured? If not measured, in "complex" in relation to what?
- o Line 442: Once again, the poorly detectable taxa argument and rarely explored environment argument needs to be reevaluated in the context of the broader literature on meromictic lakes.

Methods:

- o Line 452: Sterile containers: what kind, material, volume? How long was the water stored in these containers before it was transferred to the 2mL tubes? Was the water filtered?
- o Line 453: What volume of water was filtered for each sample?
- o Line 454: "10um plankton" – what does this mean? Is it a pre-filter through a plankton net?
- o Methods regarding the CAZYME analysis are missing from the methods.

Figures & their legends

Figure 1:

- Line 803-804: 1c and 1d: Instead of correlation, use "scatter plot" of [...]. Correlation implies the use of a statistical test to correlate variable y to x.
- Line 805: 1e. By recovered MAGS, do you mean of the final set using the hybrid method?
- Line 806: 1f. Of the MAGS of the hybrid assembly?

Figure 2:

- 2a. In the text, X new orders are identified but many more are counted here. Why?
- 2b. See comments above regarding comparing to the GTDB database.
- 2c. Note in the caption that those numbers in circle are phyla in the legend.

Figure 3.

- Note in the caption that those numbers in circle are phyla in the legend.

Figure 4.

- Rephrase the first sentence. Example: Before: "The rings from the inside to outside represent GC content (black), GC skew- (purple), GC skew + (green), coding sequence regions (blue), rRNA gene sequences (black), transfer-messenger RNA (red), and secondary metabolite gene clusters (light blue)."

After: "The rings from the inside to outside represent GC content (black), GC skew- (purple) and 831 GC skew + (green), coding sequence regions (blue). On the outmost ring are shown the rRNA gene sequences (black), transfer-messenger RNA (red), and secondary metabolite gene clusters (light blue)."

However, there are too many shades of blue (the dark blues in the rings, then at least 3 different blue shades in the outmost ring. It would be better for the reader to have more distinguishing colors. Additionally, I don't know what the dark blue in the rings mean.

Another suggestion is to add the MAG name in the figure, perhaps under the Ca. genus species name, instead of the caption only.

Supplementary Material

Figure S1:

- Figure S1. Consider spelling out BGCs.
- Line 50: Where is the glossary?

Figure S2:

- Spell out the methane gene markers

Figure S3:

- By block missing, do you mean gene missing?
- Line 167-169: Sentence unclear. How were proportions calculated?

Table S2:

- Instead of methanogenesis marker X, please spell out the name of the protein and/or function

Responses to reviewers

Manuscript number: COMMSBIO-21-1240

Salvaging complete and high-quality genomes of novel microbial species from a meromictic lake using a workflow combining long- and short-read sequencing platforms

Black represents comments to the authors from the reviewer(s)

Green represents response from the authors to the reviewer(s)

Red represents text in the revised manuscript The changes are highlighted

We thank the reviewers for their productive and insightful comments. Below, we have responded to each comment by the reviewer and, where necessary, made changes to the revised manuscript. As the indication of *communications biology*, revised text is highlighted in the main manuscript and this document. Line numbers indicate the revised version's line numbers.

Reviewers' comments:

Reviewer #1 (Remarks to the Author)

This paper describes the discovery of candidate novel microbial species through both complete or high-quality MAGs (metagenome-assembled genomes). This use of a workflow combining both short and long-read sequencing data allowed this discovery.

Overall, the paper is intelligible, well written, thorough, and describes stunning results and sturdy methodology. This workflow possesses great potential for analysis in many labs working on microbial ecology and genomics and should not be restricted to lake microbial analysis. Due to the overall quality of the paper, I do not think Major revisions are needed.

Major comments:

1. The study design and the analysis done are more than satisfactory. The results are precise and provide a glimpse of the future analysis done in microbial genomics.

I am aware of some limitations regarding the workflow, and they are well explained in the "Material and Methods" section. The restrictions I am talking about concern the

reassembly step that can worsen the bin quality and the use of N50 as a sole quality indicator. It is good to see them taken into account so clearly.

Thank you and we appreciate your comments. Indeed, the use of N50 as a sole quality indicator is not a good way. Hence, we write a script to determine whether the original or reassembled bin was better based on the value of genome completeness - 2.5 × contamination, estimated by checkM v1.1.3 as you can see in our figure 1A.

Minor comments:

1. In the "Results and Discussion," you mention the improvement of the N50 by the use of both LR and SRs instead of only the SRs for the assembly. Did you analyze the change in the completeness and contamination between the two methods? If so, you might want to include it.

Thank you for the suggestion. The results have been included.

(Line: 154-157) The median completeness of MAGs was 76.92% for SR-only and 81.26% for hybrid assembly. The median contamination was 1.61% and 1.14% for SR-only and hybrid assembly, respectively.

2. The workflow presented here is brilliant and has potentials, but only a few people will be tempted to follow it in the current state. It has been studied that reproducing a paper or reproducing the workflow presented in a paper is difficult and time-consuming (<https://doi.org/10.1371/journal.pone.0080278>, <https://doi.org/10.1371/journal.pbio.3000333>).

Have you considered making a container (Docker, Singularity, Podman) of all the tools needed for reproducing the analysis? Or have you considered making an automated pipeline with Nextflow or Snakemake that will handle the user's installation and intermediary file? Making such an automated pipeline will increase the reproducibility of

your work and the usability of the pipeline for other analyses by searchers. Such a pipeline would be a fantastic addition to the realm of tools currently available for "hybrid" metagenomics analysis. You already provide the list of tools, versions, and parameters you used. In my opinion, it is not a requirement for the publication of this paper, but providing such an automated pipeline in the future could contribute to better and more reproducible science.

Thanks for the great suggestion. We agree that the improvement can make more people to use this pipeline. According to your suggestion, we have made an automated pipeline with Snakemake and upload it to github: <https://github.com/SeanChenHCY/metaLAS>. We will maintain and update it to make it more useful for other researchers. We believe it will contribute to reproducible science. Also, a workflow with containerized conda packages will be developed.

(Line: 600)

Code availability

The workflow based on snakemake is available at <https://github.com/SeanChenHCY/metaLAS>

3. In Conclusion, this paper is an excellent paper providing exciting results on the microbial population of a meromictic lake in Siberia and a high potential workflow.

Thank you again.

Reviewer #1 comment END -----

Reviewer #2 (Remarks to the Author):

General comments:

In this paper, the authors show a proof-of-concept of how short-reads based (Illumina) and long-read based (Nanopore) sequencing technology can be combined to retrieve a high number of MAGs, including “novel” ones, using their hybrid assembly method. The method is applied to samples from Lake Shunet, a meromictic lake in Siberia. The major advantage of this hybrid method is getting higher N50, as well as completely circular MAGs, which they recovered 6 of.

Overall, the paper describes the method well, although in my major comments below, I address ways that the authors can support their claims further. In some places, the claims are insufficiently supported. Some methods are missing, which makes the reading of the paper at time unclear. I have provided some specific areas for clarification below.

The major comment is to put the results in the greater literature context. There is sufficient body of literature on both long-reads sequencing in recovering MAGs, and in MAGs from meromictic lakes and stratified lakes. Doing so would make it clearer how their study is important, and what does it contribute. It would also enable a deeper discussion of the relevance of these “novel” groups.

Thank you for your suggestion. We have added more literature about the connection between MAGs we recovered and lake systems to provide some more discussion about the role of phylogenetic groups detected in Lake Shunet.

Major comments

1. Line 183-42: Genome frequencies might not be the best indicator of whether the Sunet MAGs unique and novel, since the GTDB database comprises MAGs from many different environments. It can be hypothesized that comparing MAGs from two distinct environments, one will encounter different MAGs. A more useful comparison in terms of MAGs would be to other water lakes that encounter seasonal or annual stratification, or lakes that encounter similar environmental characteristics. For example, Cabello-Yeves et al., 2020 (ASLO L&O) has MAGs also from Siberia in Lake Baikal, and Tran et al., 2021 (ISME) has MAGs from the meromictic Lake Tanganyika. Desulfobacterota is common in hypolimnions, as are Verrucomicrobia, Bacteroidota in lakes. The authors

could put into context the MAGs that they find versus the frequency that those taxa are found in similar metagenomic datasets for example, in addition to the already existing GTDB-tk comparison.

Overall, the points about MAG novelty would be made stronger by comparing to other datasets. For example, the GEM dataset (which was mentioned in their introduction) (~52 000 MAGs) (Nayfach et al., 2020, Nature Biotechnology) could also be used to calculate the frequency of the taxa too, to see if the results about the highest proportion of X taxa remains valid.

As your suggestion, we add GEM dataset into our comparison. As the updated figure, the GEM datasets and GTDB show similar pattern. Both have highest proportions in *Proteobacteria*. The MAG collection from Shunet dataset is enriched in *Desulfobacterota*, *Verrucomicrobiota*, and *Omnitrophota*.

We also compare our results with the results from the two studies you suggested.

We can find that in these two lakes, they also recovered more *Proteobacteria* and the proportion of *Desulfobacterota* and *Omnitrophota* still higher in Lake Shunet.

We add the context into our updated manuscript:

(Line: 198-211)

The phylum frequencies differed between the genome collections of the standard database and the Shunet datasets (Fig. 2c). The GTDB and GEM mainly comprised *Proteobacteria* [1, 2]. In contrast, in genome collections from the Shunet datasets, the phylum frequency was enriched in the *Desulfobacterota*, *Verrucomicrobiota*, *Bacteroidota*, and “*Omnitrophota*.” The major phyla recovered in this study also differed from MAG studies from other ancient lakes. For example, a study also recovered MAGs from Siberia in Lake Baikal [3]. The major phyla of recovered MAGs are *Proteobacteria*, *Verrucomicrobiota*, and *Chloroflexi*. On the other hand, a 2021 metagenomics study that reconstructed MAGs from Lake Tanganyika, a freshwater meromictic lake, had higher fractions of *Proteobacteria* and *Actinobacteriota*. In both datasets, “*Margulisbacteria*,” “*Bipolaricaulota*,” and “*Caldatribacteriota*” were not seen. These results suggest that, to gain a comprehensive picture of the microbial genomes on earth, there is a strong need for future studies to explore microbiomes from various habitats, especially overlooked or understudied ones [1, 4].

In fact, the content here is not to emphasize how unique the Lake Shunet is. Here the argument we want to mention is that most of current available MAGs are recovered from animal-associated habitat or marine, freshwater (as you can see in GEM paper). Even in the GEM study, only small fraction (~10%) of MAG is from non-marine salted lake. This will skew and influence the diversity of MAGs we can understand and apprehend. Hence, it is important to recover MAGs from such rarely explored habitats to make our microbial genome encyclopedia more comprehensive. This is one of the main points in our manuscript.

2. Line 197-98: There is substantial about of microbial ecology literature on stratified lakes, including meromictic lakes, so I don't believe this statement is accurate. Overall, a more thorough screening of the water microbial ecology would allow the arguments in the paper to be put into their ecological context.

Sentence of original manuscript on Line 197-199:

“”Here we demonstrate a) the value of recovering MAGs from rarely investigated habitats to mine novel microbial function potentials and b) the advantage of combining SRs and LRs using two examples: secondary metabolite biosynthetic gene clusters (BGCs) and carbohydrate-active enzymes (CAZymes).”””

Here, we want to emphasize that we may recover novel gene from such rarely investigated habitats, instead of finding new ecological functions or roles from there MAGs. As you can see in our results in CAZymes analysis, predicted CAZyme genes from the MAGs we recovered have low protein identities with its best hits in database. The low protein identities imply these CAZyme may have novel enzymatic activity. Thank you for the comment, we have made some modification the sentence for clarification:

(Line:238-239)

Here we demonstrate a) the value of recovering MAGs from rarely investigated habitats to mine novel **function genes** and b) the advantage of combining SRs and LRs using two examples: secondary metabolite biosynthetic gene clusters (BGCs) and carbohydrate-active enzymes (CAZymes).

On the other hand, based on your suggestion, we also added more content about microbial ecology by adding discussion about the phyla of MAGs we recover and its potential role and interactions in water microbial ecology based on microbial ecology literature.

(Line:213-234)

Desulfobacterota was formally proposed as a novel phylum that includes the taxa previously classified in the class *Deltaproteobacteria* [5]. Many *Desulfobacterota* are sulfate-reducing bacteria (SRB), and play importance roles in the sulfur cycle. For

example, a study recovered numerous *Desulfobacterota* MAGs from a Siberian soda lake with complete cycling between sulfate and sulfide [6]. In some anaerobic aquatic systems, GSB formed syntrophic interactions with SRB via sulfur exchange, which were also observed in meromictic lakes such as Lake Faro and Ace Lake [7-9]. *Desulfobacterota* MAGs recovered in this study were from 5.0- and 5.5-m datasets. These two layers were dominated by purple sulfur bacteria (PSB) and green sulfur bacteria (GSB), respectively. The enrichment of recovered *Desulfobacterota* MAGs may be due to GSB having syntrophic interactions with diverse *Desulfobacterota* in the monimolimnion.

Verrucomicrobiota and “Omnitrophota” belong to the PVC group, and both were found and proposed recently. *Verrucomicrobiota* are abundant and ubiquitous in various soil and water systems. Although they have received more attention recently, only a few of them have been isolated, and their functions and ecophysologies in water systems are not widely understood. A study in four Swedish lakes showed that the *Verrucomicrobiota* are associated with cyanobacterial blooms [10]. On the other hand, many studies showed that *Verrucomicrobia* contain higher proportions of carbohydrate-active enzymes-related genes [11, 12] and can digest complex polysaccharides for growth [13]. *Verrucomicrobia* may also serve as important (poly)saccharide degraders in Lake Shunet.

Minor comments:

1. Line 80: ref needed

Corrections have been made per your suggestion. (Line: 63)

2. Line 109: using different binning methods on the short reads set, are more high-quality MAGs recovered? Using multiple binning softwares, followed by dereplication of the MAGs, can often increase the number of MAGs and the MAG quality. Additionally, for novel bacteria/archaea belonging to the DPANN or CPR, usual core-gene set such as those in CheckM might not capture their true completeness.

Yes, when we used multiple binning softwares, followed by dereplication (as we did in our workflow). The number of high-quality (completeness > 90, contamination < 5) MAG from short reads increases to 44. However, we recovered 65 high-quality MAGs by our

hybrid assembly with reassembly. The median completeness of MAGs from SR-only is 76.92% but from hybrid assembly is 81.26%. The median contamination is 1.61% and 1.14% for SR-only and hybrid assembly, respectively (As below figure).

For

the second issue, we do admit that completeness evaluation based on lineage-specific markers (as checkM does) has its limitation. It depends on the comprehensiveness of our current database. Novel bacteria/archaea may carry multiple copy of markers or lack of consensus marker genes, which will influence the quality evaluation. However, currently it is the best or said a better way to evaluate the completeness of metagenome-assembled genomes. Hence, it is importance to explore these novel lineages and identify their lineage-specific markers to expand our current database. Long-read technique may provide the opportunity to do so. Using Long-read method, we have more chance to recover complete circular genomes. If one contig is labeled as circular but its completeness evaluated by checkM is abnormal, and its phylogenetic placement is novel, we can elucidate that the contig may belong to a novel species which has unusual marker-gene profile.

3. Line 128: More methodological details are needed to describe how the SRs were used to correct the contigs generated by the LR method, such that would enable the reader to replicate their experiment.

We use pilon to correct the contigs generated by the LR method. The details were described in method section. However, according to your kind suggestion, we rephrased this part (Line 128 in original version) for clarification.

(Line:154-157) Hence, the contigs generated by LRs were polished using pilon [14] with SRs from Illumina sequencing. The SRs were first mapped to the assemblies from LRs with BWA-MEM [15]. Sequentially, pilon was used to automatically evaluate the read alignments to identify the disagreement between assemblies and SRs and makes

corrections to fix base errors and small indels based on the evidence from alignments weighted by base coverage and quality of the SRs.

4. Line 156: GTDB / GTDB-tk requires a citation.

Corrections have been made.

(Line: 151)

5. Line 170: De-replicated instead of de-duplicated

Corrections have been made per your suggestion.

(Line: 165)we clustered and de-replicated the genomes.....

6. Line 182: Poorly and unusual characterized phyla – in comparison to what? Is it in comparison to other water lakes, or else?

In comparison the common phyla, such as *Proteobacteria*, *Firmicutes*, *Actinobacteria*, *Bacteroidetes*, and *Cyanobacteria*, in current bacterial genome database. For example, as of 2021/7/1, there are 40,718 *Proteobacteria* genomes, 21,138 *Firmicutes* genomes, 13,246 *Actinobacteria* genomes, 4,600 *Bacteroidetes* genomes, and 1,653 *Cyanobacteria* genomes in Integrated Microbial Genomes (IMG) database. However, there are only 34 *Armatimonadetes*, 15 *Margulisbacteria*, 17 *Bipolaricaulota*, 60 *Cloacimonadota* genomes in IMG. Moreover, currently, there is no bacterial isolate for *Margulisbacteria*, *Bipolaricaulota* and, *Cloacimonadota* (but there are 7 *Armatimonadetes* isolates). Genomes in these phyla are also scarce in other database such as GTDB and GEM dataset (figure 2b)(also see below text). The characteristics and diversity of these uncultured bacteria are not as clear as the common phyla mentioned above. Hence, these bacteria are also called microbial dark matters. Currently, the metabolic features of the species under these phyla are inferred by culture-independent methods such as metagenomics and metatranscriptomics. However, more genomes in these phyla are needed in order to identify the common features and diversity of these phyla by comparative genomics.

Here, we reword the sentence for clarification.

(Line: 181-197)

The MAGs were distributed across 24 phyla, and 11 MAGs belonged to *Candidatus* *Patescibacteria* (also known as Candidate Phyla Radiation, CPR), so-called microbial

dark matter because not enough is known about their biology. Recovered MAGs also included “Margulisbacteria,” “Bipolaricaulota,” “Cloacimonadota,” and “Caldatribacteriota,” phyla that were novel and poorly-characterized in the current bacterial genome databases GTDB and IMG compared to the common phyla, such as *Proteobacteria*, *Firmicutes*, *Actinobacteria*, *Bacteroidetes*, and *Cyanobacteria*. “Margulisbacteria” was first identified in 2016 from metagenomes of groundwater and sediment [16]. “Bipolaricaulota” (Previously known as “Acetothermia”) was first recovered from the metagenome of the thermophilic microbial mat community in 2012 [17]. “Cloacimonadota” (“Cloacimonetes”) was first described in 2008 from anaerobic digesters [18]. These phyla have not yet been cultivated, except the first “Caldatribacteriota” isolate published in 2020 [19]. There only are 37 “Margulisbacteria,” 21 “Bipolaricaulota,” 27 “Cloacimonadota,” and 19 “Caldatribacteriota” species representative genomes in GTDB-r95. Our newly recovered MAGs of these uncultivated phyla will be useful tools for exploring these phyla.

7. Line 204: What makes these metabolites novel?

Thank you for the comment. We agree that the description is not clearly defined. Microbes produce diverse secondary metabolites. These secondary metabolites are diverse in terms of their structures and functions. Identifying functionally and structurally novel metabolites may provide new insights in ecological interactions and have implication in drug development. To make the sentence clearer, we add “functionally and structurally” to the sentence.

(Line: 245)

Identifying **structurally and functionally** novel secondary metabolites enables us to understand the ecological interactions among the microbes.

8. Line 208: Where are the 140 MAGs from? On line 173, the previous mention of the number of MAGs was 165.

Among the total 165 MAGs, we can only identify BGCs in the 140 MAGs, and we cannot detect BGCs in the other 25 MAGs, so the number is $165 - 25 = 140$

For clarification, we rephrase the sentence

(Line: 249)

In our MAG collection, we identified 414 putative BGCs from 140 MAGs out of a total 165 recovered MAGs (Fig. S1a).

9. Line 227: Instead of “on the other hand”, “additionally” might be better to use because it would Line 242: Overuse of the novelty aspect but without support.

Thank you. We have corrected it.

(Line: 269)

10. Line 247: What criteria were used to determine novelty? The authors mention bootstrap support values. I see that %ANI to nearby genomes is provided for other groups later on in the manuscript, but it would be helpful to provide it here too.

We used GTDB-tk to determine novelty, which used 95% ANI for species boundary. However, for higher ranks, GTDB-tk use relative evolutionary divergence (RED) value and phylogenetic placement to determine the novelty. The RED of an internal node in a tree is determined by $p + (d/u) \times (1 - p)$ where :

p = RED of its parent

d = branch length to its parent

u = average branch length from the parent node to all extant taxa descendant from n .

The more detail on the classification by GTDB-Tk based on RED can be found in their manuscript.

The RED values of novel genomes we recovered are provided in supplementary dataset.

For clarification, we rephrase the sentence:

(Line: 289)

From the 5.5 m dataset, we identified two MAGs belonging to candidate novel families under *Methanomassiliicoccales* and *Iainarchaeales* (MAG ID: M55A1 and M55A2, respectively) based on the relative evolutionary divergence (RED) and phylogenetic placement determined by GTDB-Tk (Dataset S2), and one MAG belonging to a potential novel species under Nanoarchaeota based on a 95% ANI cutoff for species boundary (Dataset S2).

Also, we add the GTDB-Tk taxonomy classification in method section.:

(Line: 586-589)

Taxonomic assignment was performed by GTDB-Tk v1.3.0 [70]. GTDB-Tk classifies bacterial and archaeal genomes and identifies novel taxa by determining the phylogenetic placement and relative evolutionary divergence (RED) values of query genomes in the GTDB reference tree and using a 95% ANI cutoff for species boundary.

11. Line 268-269: Have these Methanomassiliicoccales been found in other lakes?

Yes, according to 16S rRNA survey, environmental *Methanomassiliicoccales* can be found in meromictic lake, hot springs, river sediments, muds, etc. [20]

12. Line 297: What membrane pore size was used?

We used 10- μ m plankton nets to filter large particle and 0.2- μ m to remove the fraction of virus-like particles. The sentence has been corrected.

(Line: 339)

Our samples were collected from saline water, and 10- μ m plankton nets were used to filter large organisms.

13. Line 300: Contamination can be a result of binning, but is not always the case using binning. Additionally, it is possible (although not common) to retrieve fully circular MAGS from binning using short-reads. I suggest this phrasing to be more accurate (suggestions only, authors can reword as they see fit): “Leveraging long-reads sequencing along with short-read sequencing to improve MAG reconstruction, can result in comprehensive genomes with less genome contamination and binning errors”.

The sentence has been revised per your suggestion.

(Line: 342)

Leveraging long- and short-read sequencing together to improve MAG reconstruction can result in comprehensive genomes with less genome contamination and fewer binning errors.

14. Line 309: What are these “extreme” cold environments? Please mention the temperature, general climate, salinity levels.

The temperature and climate have been added in the paragraph as your suggestion. The salinity levels have been added in sampling information in materials & methods.

(Line: 351-353) The water temperature of samples in the mixolimnion we collected in July 2010 ranges from 6.5 to 15.5°C [21, 22]. In winter, ice cover the lake surface, and the temperature is below 0°C down to 4 m deep [23].

(Line: 513-516) The physicochemical properties of samples were mentioned in a previous study [21]. The pH was 8.1, 7.6, and 6.7; water temperature was 15.5, 9.5, and 7.5°C; and salinity was 26, 40, 71g/L in the 3.0-, 5.0-, and 5.5-m samples, respectively.

15. Line 317: What is the pH in this environment?

The pH in 3.0m layer is 8.1 when we collected the sample [21]. However, please note that *Simkaniaceae*, like *Chlamydia*, are obligate intracellular bacteria that live in host cells. Hence, GAD may help *Simkaniaceae* to adapt to not only acidic external, but also host environments. For example, a study found that *Francisella*, an intracellular bacterium, can use GAD to tolerate host acid environments such as acidified phagosomal compartment [24]. Therefore, we rephrased the sentence:

(Line:364) can also be found in the two novel *Simkaniaceae* genomes, indicating that the bacteria can use the system to tolerate acidic external or host intracellular environments.

16. Line 318-324: The argument here is that the metabolism of these two mags differs, but the only difference listed is the assimilatory sulfate reduction pathway, and the sulfate permease gene. How do other aspects of the sulfur cycle (the authors' example) differ across genomes? Do they differ in other aspects of metabolism than sulfur cycling? More evidence is needed to support that it "broadens our knowledge of the metabolic versatility in *Simkaniaceae*" (example: Line 324, 351).

Indeed, more evidence is needed to support that statement. In our analysis, we only identify the difference in assimilatory sulfate reduction pathway in these two new *Simkaniaceae*. Hence, we removed the statement:"broadens our knowledge of the metabolic versatility in *Simkaniaceae*"

17. Line 325 and elsewhere: Do the abbreviated MAG names (e.g. M30B3) signify anything? (e.g. sample provenance, metadata, etc.).

The number between M and B represent the depth which the bin recovered from. For

example, M30B3 was recovered from 3.0m and M50B4 was recovered from 5.0m. M55B102 was from 5.5m. The number after B (M30B3) is given randomly.

18. Line 370-371: Which vesicle protein genes were found?

We found two copies of *gvpA*, a gene encoding gas vesicle structure protein, in these genomes. The gene name has been added for clarification.

(Line: 416-417) Our genomic analysis revealed that M50B4 contained two copies of the gas vesicle structure protein gene *gvpA*.

19. Line 375: Provide clarification of how the environmental conditions are changing, and how the adaptations are possibly related to these conditions. For example, are there big changes in terms of light or oxygen, as cited in line 374-375?

The depth of chemocline fluctuates between 4.5 to 5.5m and we can observe the corresponding shift of PSB. Gas vesicles may help M50B4 to move along with the change in chemocline. On the other hand, the M50B4 was also found in Lake Shira according to 16S rRNA analysis. Lake Shira is characterized by less stable stratification. Previous study recorded that redox transition zone shifted upwards in two days and the PSBs are able to maintain its position near the zone by shifting upwards. Hence, gas vesicles may play a role in rapid environmental changes.

We have added these examples into our text.

(Line: 421-431) Gas vesicles enable *T. sp.* M50B4 cells to modulate their buoyancy so they can move to the locations with optimal conditions for anoxygenic photosynthesis [25]. For example, the position of the chemocline, a zone with a H₂S concentration and illumination that are optimal for anoxygenic photosynthesis, fluctuated between 4.5 and 5.5 m deep from 2003 to 2009. During the period, the corresponding change in the PSB peak location was also observed, indicating that gas vesicles may help M50B4 move along with the location change in the chemocline [23]. On the other hand, 16S rRNA analysis showed that M50B4 can also be found in Lake Shira, a less stable meromictic lake near Lake Shunet. In summer 2004, the redox zone of Lake Shira was recorded to have shifted upwards in two days [26]. This showed that the PSB are able to maintain

their position near the zone by shifting upwards, indicating that gas vesicles may help M50B4 deal with rapid environmental changes.

20. Line 388: While the organisms carry the genes, they may not utilize them year-round. Perhaps change for “carry the genes for DMS metabolism”.

Thank you for the suggestion. We corrected the sentence. (Line: 442)

21. Line 388: What is meant by “extremely dense”? What criteria is used? Also check and remove superfluous words such as “extreme” (line 388, 87, 309, 344) when no further explanations of how or why they are extreme are provided.

The words on these lines have been removed per your suggestion. Also, we provide the cell density of purple sulfur bacteria at 5.0m into the text.

(Line: 390-393)

Lake Shunet features dense purple sulfur bacteria (PSB) in its chemocline (5.0 m) layer ($>10^8$ cells/mL), and the density is comparable to that of Lake Mahoney (Canada), renowned for containing the most PSB of any lake in the world (4×10^8 cells/mL)

22. Line 432: how is microbial community complexity measured? If not measured, in “complex” in relation to what?

We have determined the bacterial diversity by Shannon index (H') based on 16S rRNA and whole genome shotgun (WGS) data. The Shannon index based on 16S rRNA (97% OTU) is 5.46 which is higher than those in other aquatic ecosystems, including two hypersaline meromictic, the Ursu Lake (bacteria $H' = 4.64-5.45$) and Fara Fund Lake (bacteria $H' = 3.74$) [27], freshwater reservoirs (bacteria $H' = 5.34$) [28]. Based on WGS, the Shannon index ranges from 7.67–7.81 in Lake Shunet, which is higher than Shannon index of 6.46-6.93 in North Pacific Ocean [29]. We also compared microbial diversity and richness in Lake Shira, Lake Shunet, and Lake Oigon, and Lake Shunet show highest microbial diversity and richness [22].

For clarification. The sentence has been reworded as follow:

(Line: 485).....from water samples of Lake Shunet, a meromictic lake with a diverse microbial community.

We also provided more information in introduction

(Line: 89-92)..... Our previous 16 rRNA amplicon survey showed that the lake contains diverse microbial communities with a higher Shannon diversity index and Chao1 richness estimator than Lake Shira and Lake Oigon, two another saline meromictic lakes near the center of Asia [22]. More importantly,....

23. Line 442: Once again, the poorly detectable taxa argument and rarely explored environment argument needs to be reevaluated in the context of the broader literature on meromictic lakes.

As our response to major comment #1 and minor comment# 6.

“Margulisbacteria”, “Bipolaricaulota“, “Cloacimonadota“, and “Caldatribacteriota“, novel and poorly-characterized phyla in comparison the common phyla, such as *Proteobacteria*, *Firmicutes*, *Actinobacteria*, *Bacteroidetes*, and *Cyanobacteria* in current bacterial genome database GTDB and IMG. These bacteria are also called microbial dark matters because they are uncultivated. Currently, the metabolic features of the species under these phyla are inferred by culture-independent methods.

On the other hand, most of current available MAGs are recovered from animal-associated habitat or marine, freshwater (as you can see in GEM paper). Even in GEM study, only small fraction (~10%) of MAG is from non-marine salted lake. This will skew and influence the diversity of MAGs we obtain. Hence, it is important to recover MAGs from such rarely explored habitats to make our microbial genome encyclopedia more comprehensive.

Methods:

24. Line 452: Sterile containers: what kind, material, volume? How long was the water stored in these containers before it was transferred to the 2mL tubes? Was the water filtered? We used sterile 5 L polypropylene bottles to collect water. The water was store for 3 hours before transferred into the 2mL tubes. The water was filtered with 10- μ m plankton net. 20L water was filtered for each sample. The information has been added (Line: 505-509) Briefly, 20 L of water were pumped from each depth into four sterile 5-L polypropylene bottles. Water was filtered through 10- μ m plankton net to remove large particles. The water was stored for 3 hours in the bottles, and part of the water was then

transferred into sterile 2.0-ml screw tubes (SSlbio®) and stored at -80°C until DNA extraction

25. Line 453: What volume of water was filtered for each sample?

20L water was filtered for each sample. The information has been added

(Line: 505) Briefly, 20 L of water were pumped from each depth into four sterile 5-L polypropylene bottles.....

26. Line 454: “10um plankton” – what does this mean? Is it a pre-filter through a plankton net?

Yes, 10-µm plankton nets were used for pre-filter to remove large particles. The information has been added

(Line: 506) ...Water was filtered through 10-µm plankton net to remove large particles.....

27. Methods regarding the CAZYME analysis are missing from the methods.

The CAZYME analysis method was added

(Line: 579-583) Carbohydrate-Active Enzymes (CAZymes) in MAGs were predicted and classified using run_dbcan v2 by HMMER search against the dbCAN HMM (hidden Markov model) v9 database [30]. The putative CAZymes were blasted against the NCBI nr database using BLASTp [31] to search their closest homologs and determine the protein identities between predicted CAZymes and their homologs.

Figures & their legends

28. Figure 1:

- Line 803-804: 1c and 1d: Instead of correlation, use “scatter plot” of [...]. Correlation implies the use statistical test to correlate variable y to x.

Thank you for the comment. Corrections have been made per your suggestion.

(Line: 943, 944)

- Line 805: 1e. By recovered MAGS, do you mean of the final set using the hybrid method?

Yes, the plots indicate the completeness and contamination of MAGs recovered from hybrid method with reassembly. The information has been added.

(Line: 946) e. The completeness and contamination of MAGs recovered from the hybrid method with reassembly.

- Line 806: 1f. Of the MAGS of the hybrid assembly?

Yes, the information has been added.

(Line: 947) f. Venn diagram from the ratio of MAGs of the hybrid assembly

29. Figure 2:

- 2a. In the text, X new orders are identified but many more are counted here. Why?

In accord with the text, six potential novel orders are identified.

- 2b. See comments above regarding comparing to the GTDB database.

As our response for major comment#1

- 2c. Note in the caption that those numbers in circle are phyla in the legend.

The information has been added per your suggestion. (Line: 959)

30. Figure 3.

- Note in the caption that those numbers in circle are phyla in the legend.

The information has been added per your suggestion. (Line: 968)

31. Figure 4.

- Rephrase the first sentence. Example: Before: "The rings from the inside to outside represent GC content (black), GC skew- (purple), GC skew + (green), coding sequence regions (blue), rRNA gene sequences (black), transfer-messenger RNA (red), and secondary metabolite gene clusters (light blue)." After: "The rings from the inside to outside represent GC content (black), GC skew- (purple) and 831 GC skew + (green), coding sequence regions (blue). On the outmost ring are shown the rRNA

gene sequences (black), transfer-messenger RNA (red), and secondary metabolite gene clusters (light blue).”

The sentences have been rephrased according to your suggestion:

(Line:972-976) The rings from the inside to outside represent GC content (black), GC skew- (purple), and GC skew + (green). The next two blue rings represent coding sequences on the forward and reverse strands, respectively. On the outmost ring are the rRNA gene sequences (black), transfer-messenger RNA (red), and secondary metabolite gene clusters (yellow).

-
- However, there are too many shades of blue (the dark blues in the rings, then at least 3 different blue shades in the outmost ring. It would be better for the reader to have more distinguishing colors. Additionally, I don't know what the dark blue in the rings mean.

According to your suggestion, we changed the light blue with yellow to represent secondary metabolite gene clusters to make the figure clearer. On the other hand, two blue rings are used to represent coding sequence regions. The inner blue ring represents CDS on forward strand and the other blue ring represents CDS on reverse strand. The information has been added in the legend:

(Line: 973) The next two blue rings represent coding sequences on the forward and reverse strands, respectively.....

On the other hand, the legend is added on the upper left corner.

- Another suggestion is to add the MAG name in the figure, perhaps under the Ca. genus species name, instead of the caption only.

Changes have been made per your suggestion. We add the MAG IDs under the Ca. genus species name

32. Supplementary Material

33. Figure S1:

- Figure S1. Consider spelling out BGCs.

Full names of some BGCs are so long that can make the figure messy. Hence, we add the full names in the legend:

(Line: 56-61) Abbreviations in figure S1a: T3PKS, Type III Polyketide synthase (PKS); NPRS, Non-ribosomal peptide synthetase cluster; hglE-KS, heterocyst glycolipid synthase-like PKS; LAP, Linear azol(in)e-containing peptides; NAGGN, N-acetylglutaminyglutamine amide; head_to_tail, Head-to-tail cyclised cluster; acyl_amino_acids, N-acyl amino acid cluster; TfuA-related; TfuA-related ribosomally synthesized and post-translationally modified peptides. Abbreviation in figure S2b: c:

- Line 50: Where is the glossary?

The glossary can be found in antiSmash online document. The website link has been added in the caption.

(Line 50-51) ...can be found in the glossary from antiSMASH (<https://docs.antismash.secondarymetabolites.org/glossary/>)

34. Figure S2:

- Spell out the methane gene markers

methanogenesis marker protein (mmp) 1 to 17 are putative protein families that were identified using phylogenetic profiling [32]; these markers are universal and common in the archaeal methanogens. Apart from the key functionally markers (*mcrA*, *mcrB*, *mcrG*), mmPs are used in several studies about archaeal methanogens [33-35]. The exact functions of mmPs are unknown, except mmp10, but are thought to be linked to methanogenesis. Being functionally unknown markers, there are no formal gene name for these markers. However, we do realize that some researcher use “mmp X” to represent these methanogenesis marker proteins and coding genes [36, 37]. Hence, we change the original label with *mmp1* - *mmp17*. Also, we define the abbreviation in the caption.

(Line: 93) Abbreviation: mmp, methanogenesis marker protein.

35. Figure S3:

- By block missing, do you mean gene missing?

Sorry for the confusion. It is not “gene missing”. KEGG modules are consisted of multiple building blocks. Each building block may comprise more than two genes. In this case, the blocking is labeled as missing, if one of genes cannot be found. Besides,

individual block may also have alternative genes (Genetic redundancy). The below is the example.

Example:

Entry M00012
 Name Glyoxylate cycle
 Definition K01647 (K01681,K01682) K01637 (K01638,K19282) (K00026,K00025,K00024)

This module contains 5 blocks. For the block A. The block is present if gene K01647 can be found. However, for block B, as long as K00026 or K00025 is present, the block is present.

In order to make the figure clearer, we added explanation in the caption.

(Line: 128-130) Each KEGG module is made of multiple functional units (blocks). Missing blocks represents missing gene(s) that form a functional unit. Incomplete modules indicate that more than two blocks are missing.

36. Figure S4:

- Line 167-169: Sentence unclear. How were proportions calculated?

Taking flagella gene in genome X as an example:

$$\text{Proportion} = \frac{\text{The number of flagellar genes that can be indentified in genome X}}{\text{Total number of flagellar gene in KEGG database}}$$

The sentence has been reworded for clarification.

(Line: 171-174) The heatmap represents the proportions calculated by dividing the number of flagella, chemotaxis, and pilus-related genes found in the genome with related gene numbers in the KEGG database (list in Table S2)

37. Table S2:

- Instead of methanogenesis marker X, please spell out the name of the protein and/or function.

As our response to point 34, the methanogenesis marker proteins are functionally unknown except methanogenesis marker protein 10 (mmp10). Hence, we only add the putative function of mmp10 into our table.

Reviewer #2 END -----

1. Nayfach, S., et al., *A genomic catalog of Earth's microbiomes*. Nat Biotechnol, 2020.
2. Parks, D.H., et al., *Recovery of nearly 8,000 metagenome-assembled genomes substantially expands the tree of life*. Nature Microbiology, 2017. **2**(11): p. 1533-1542.
3. Cabello-Yeves, P.J., et al., *Microbiome of the deep Lake Baikal, a unique oxic bathypelagic habitat*. Limnology and Oceanography, 2020. **65**(7): p. 1471-1488.
4. Obbels, D., et al., *Bacterial and eukaryotic biodiversity patterns in terrestrial and aquatic habitats in the Sor Rondane Mountains, Dronning Maud Land, East Antarctica*. Fems Microbiology Ecology, 2016. **92**(6).
5. Waite, D.W., et al., *Proposal to reclassify the proteobacterial classes Deltaproteobacteria and Oligoflexia, and the phylum Thermodesulfobacteria into four phyla reflecting major functional capabilities*. International Journal of Systematic and Evolutionary Microbiology, 2020. **70**(11): p. 5972-6016.
6. Vavourakis, C.D., et al., *Metagenomes and metatranscriptomes shed new light on the microbial-mediated sulfur cycle in a Siberian soda lake*. BMC Biology, 2019. **17**(1).
7. Ng, C., et al., *Metaproteogenomic analysis of a dominant green sulfur bacterium from Ace Lake, Antarctica*. ISME J, 2010. **4**(8): p. 1002-19.
8. Lentini, V., C. Gugliandolo, and T.L. Maugeri, *Vertical distribution of Archaea and Bacteria in a meromictic lake as determined by fluorescent in situ hybridization*. Curr Microbiol, 2012. **64**(1): p. 66-74.
9. Lauro, F.M., et al., *An integrative study of a meromictic lake ecosystem in Antarctica*. Isme Journal, 2011. **5**(5): p. 879-895.
10. Eiler, A. and S. Bertilsson, *Composition of freshwater bacterial communities associated with cyanobacterial blooms in four Swedish lakes*. Environ Microbiol, 2004. **6**(12): p. 1228-43.
11. Cabello-Yeves, P.J., et al., *Reconstruction of Diverse Verrucomicrobial Genomes from*

- Metagenome Datasets of Freshwater Reservoirs*. Front Microbiol, 2017. **8**: p. 2131.
12. He, S., et al., *Ecophysiology of Freshwater Verrucomicrobia Inferred from Metagenome-Assembled Genomes*. mSphere, 2017. **2**(5).
 13. Sichert, A., et al., *Verrucomicrobia use hundreds of enzymes to digest the algal polysaccharide fucoidan*. Nat Microbiol, 2020. **5**(8): p. 1026-1039.
 14. Walker, B.J., et al., *Pilon: an integrated tool for comprehensive microbial variant detection and genome assembly improvement*. PLoS One, 2014. **9**(11): p. e112963.
 15. Li, H. and R. Durbin, *Fast and accurate short read alignment with Burrows-Wheeler transform*. Bioinformatics, 2009. **25**(14): p. 1754-60.
 16. Anantharaman, K., et al., *Thousands of microbial genomes shed light on interconnected biogeochemical processes in an aquifer system*. Nat Commun, 2016. **7**: p. 13219.
 17. Takami, H., et al., *A deeply branching thermophilic bacterium with an ancient acetyl-CoA pathway dominates a subsurface ecosystem*. PLoS One, 2012. **7**(1): p. e30559.
 18. Pelletier, E., et al., *"Candidatus Cloacamonas acidaminovorans": genome sequence reconstruction provides a first glimpse of a new bacterial division*. J Bacteriol, 2008. **190**(7): p. 2572-9.
 19. Katayama, T., et al., *Isolation of a member of the candidate phylum 'Atribacteria' reveals a unique cell membrane structure*. Nat Commun, 2020. **11**(1): p. 6381.
 20. Cozannet, M., et al., *New Insights into the Ecology and Physiology of Methanomassiliicoccales from Terrestrial and Aquatic Environments*. Microorganisms, 2021. **9**(1).
 21. Wu, Y.T., et al., *Comprehensive Insights Into Composition, Metabolic Potentials, and Interactions Among Archaeal, Bacterial, and Viral Assemblages in Meromictic Lake Shunet in Siberia*. Frontiers in Microbiology, 2018. **9**.
 22. Baatar, B., et al., *Bacterial Communities of Three Saline Meromictic Lakes in Central Asia*. Plos One, 2016. **11**(3).
 23. Rogozin, D., V.V. Zykov, and A.G. Degermendzhi, *Ecology of the purple sulfur bacteria in the highly stratified meromictic lake Shunet (Siberia, Khakasia) in 2002-2009*. Mikrobiologiya, 2012. **81**(6): p. 786-95.
 24. Ramond, E., et al., *Glutamate Utilization Couples Oxidative Stress Defense and the Tricarboxylic Acid Cycle in Francisella Phagosomal Escape*. Plos Pathogens, 2014. **10**(1).

25. Walsby, A.E., *Gas vesicles*. Microbiol Rev, 1994. **58**(1): p. 94-144.
26. Rogozin, D.Y., V.V. Zykova, and M.O. Tarnovskii, *Dynamics of Purple Sulfur Bacteria in a Meromictic Saline Lake Shunet (Khakassia, Siberia) in 2007-2013*. Mikrobiologiya, 2016. **85**(1): p. 73-82.
27. Andrei, A.S., et al., *Contrasting taxonomic stratification of microbial communities in two hypersaline meromictic lakes*. Isme Journal, 2015. **9**(12): p. 2642-2656.
28. Tseng, C.H., et al., *Microbial and viral metagenomes of a subtropical freshwater reservoir subject to climatic disturbances*. Isme Journal, 2013. **7**(12): p. 2374-2386.
29. DeLong, E.F., et al., *Community genomics among stratified microbial assemblages in the ocean's interior*. Science, 2006. **311**(5760): p. 496-503.
30. Zhang, H., et al., *dbCAN2: a meta server for automated carbohydrate-active enzyme annotation*. Nucleic Acids Research, 2018. **46**(W1): p. W95-W101.
31. Camacho, C., et al., *BLAST+: architecture and applications*. BMC Bioinformatics, 2009. **10**: p. 421.
32. Basu, M.K., J.D. Selengut, and D.H. Haft, *ProPhylo: partial phylogenetic profiling to guide protein family construction and assignment of biological process*. BMC Bioinformatics, 2011. **12**: p. 434.
33. Zinke, L.A., et al., *Evidence for non-methanogenic metabolisms in globally distributed archaeal clades basal to the Methanomassiliicoccales*. Environ Microbiol, 2021. **23**(1): p. 340-357.
34. Hua, Z.S., et al., *Insights into the ecological roles and evolution of methyl-coenzyme M reductase-containing hot spring Archaea*. Nat Commun, 2019. **10**(1): p. 4574.
35. Li, Y., et al., *The complete genome sequence of the methanogenic archaeon ISO4-H5 provides insights into the methylotrophic lifestyle of a ruminal representative of the Methanomassiliicoccales*. Stand Genomic Sci, 2016. **11**(1): p. 59.
36. Radle, M.I., et al., *Methanogenesis marker protein 10 (Mmp10) from Methanosarcina acetivorans is a radical S-adenosylmethionine methylase that unexpectedly requires cobalamin*. J Biol Chem, 2019. **294**(31): p. 11712-11725.
37. Kelly, W.J., et al., *Complete Genome Sequence of Methanogenic Archaeon ISO4-G1, a Member of the Methanomassiliicoccales, Isolated from a Sheep Rumen*. Genome Announc, 2016. **4**(2).

REVIEWERS' COMMENTS:

Reviewer #2 (Remarks to the Author):

Dear authors,

Thank you for taking the time to carefully respond to my comments. The responses are clear and thorough. First, I particularly appreciate your time to make the snakemake pipeline which contributes to reproducible science and will allow readers in the future to also use the LR-SR method to assemble highly complete and circular genomes. Second, thank you for the clarification about the main argument of your study. The emphasis on non-marine salty lakes is now much clearer. Finally, I also appreciate that you included more ecological context.

Overall, the authors have addressed all my comments in a satisfactory manner.

Responses to reviewers

Manuscript number: COMMSBIO-21-1240A

Salvaging complete and high-quality genomes of novel microbial species from a meromictic lake using a workflow combining long- and short-read sequencing platforms

Black represents comments to the authors from the reviewer(s)

Green represents response from the authors to the reviewer(s)

REVIEWERS' COMMENTS:

Reviewer #2 (Remarks to the Author):

Dear authors,

Thank you for taking the time to carefully respond to my comments. The responses are clear and thorough. First, I particularly appreciate your time to make the snakemake pipeline which contributes to reproducible science and will allow readers in the future to also use the LR-SR method to assemble highly complete and circular genomes. Second, thank you for the clarification about the main argument of your study. The emphasis on non-marine salty lakes is now much clearer. Finally, I also appreciate that you included more ecological context.

Overall, the authors have addressed all my comments in a satisfactory manner.

We thank you for appreciating our work and your previous suggestions and comments that improved our work.